# SCRWKV: Ultra-Compact Structure-Calibrated Vision-RWKV for Topological Crack Segmentation

**Hanxu Zhang** [1 2]  **Chen Jia** [1 2]  **Hui Liu** [1 2]  **Xu Cheng** [1 2]  **Fan Shi** [1 2]  **Shengyong Chen** [1 2]

## Abstract

Achieving pixel-level accurate segmentation of structural cracks across diverse scenarios remains a formidable challenge. Existing methods face significant bottlenecks in balancing crack topology modeling with computational efficiency, often failing to reconcile high segmentation quality with low resource demands. To address these limitations, we propose the Ultra-Compact Structure-Calibrated Vision RWKV (SCRWKV), a network that achieves high-precision modeling via a novel Structure-Field Encoder (SFE) backbone while maintaining linear complexity. The SFE integrates the Adaptive Multi-scale Cascaded Modulator (AMCM) to enhance texture representation and utilizes the Structure-Calibrated Insight Unit (SCIU) as its core engine. Specifically, the SCIU employs the Geometry-guided Bidirectional Structure Transformation (GBST) to capture topological correlations and integrates the Dynamic Self-Calibrating Decay (DSCD) into Dy-WKV to suppress noise propagation. Furthermore, we introduce a lightweight Cross-Scale Harmonic Fusion (CSHF) decoder to achieve precise feature aggregation. Systematic evaluations on multiple benchmarks characterized by complex textures and severe interference demonstrate that SCRWKV, with only 1.22M parameters, significantly outperforms SOTA methods. Achieving an F1 score of 0.8428 and mIoU of 0.8512 on the TUT dataset, the model confirms its robust potential for efficient real-world deployment. The code is available at https://github.com/zhxhzy/SCRWKV.

[1]Engineering Research Center of Learning-Based Intelligent System, Ministry of Education, Tianjin University of Technology, Tianjin 300384, China [2]Key Laboratory of Computer Vision and System, Ministry of Education, Tianjin University of Technology, Tianjin 300384, China. Correspondence to: Xu Cheng <xu.cheng@ieee.org>.

*Proceedings of the 43$^{rd}$ International Conference on Machine Learning*, Seoul, South Korea. PMLR 306, 2026. Copyright 2026 by the author(s).

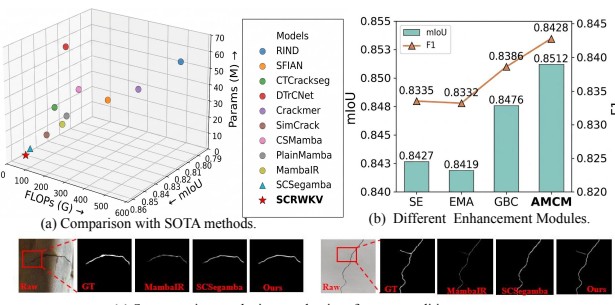

*Figure 1.* Performance of SCRWKV on multi-scenario TUT (Liu et al., 2024a) dataset. (a) Comparison with SOTA methods. (b) Impact of different enhancement modules on performance. (c) Visual results under complex interference.

## 1  Introduction

Under long-term loading and environmental disturbances, asphalt pavements, concrete components, and metal structures are highly susceptible to cracking (Chen et al., 2022; 2024; Hsieh & Tsai, 2020; Kheradmandi & Mehranfar, 2022; Liao et al., 2022). While precise segmentation is pivotal for ensuring safety, the extreme morphological diversity and severe background noise in real-world scenarios (Lang et al., 2024) pose a formidable challenge to achieving high-precision segmentation on resource-constrained edge devices (Liao et al., 2022).

Existing models struggle to strike a balance between efficiency and structural modeling capability. CNN-based methods such as SFIAN (Cheng et al., 2023) and ECSNet (Zhang et al., 2023) effectively utilize local inductive biases but are restricted by limited receptive fields, often resulting in segmentation fractures when processing long-range cracks. Transformer-based approaches like DTrCNet (Xiang et al., 2023), MFAFNet (Dong et al., 2024), and Crackmer (Wang et al., 2024) excel at capturing global dependencies, yet their quadratic computational complexity hinders real-time application. Recently, Selective State Space Models (SSMs) have garnered significant attention due to their linear complexity, with Vision Mamba (ViM) (Zhu et al., 2024) extending the Mamba architecture to the visual domain. As shown in Figure 1(a), although methods such as CSMamba (Liu et al., 2024b), MambaIR (Guo et al., 2024), PlainMamba (Yang et al., 2024), and SCSegamba (Liu et al., 2025) have achieved certain progress in reducing

visual modeling complexity by introducing SSMs, their core limitations regarding structural crack segmentation tasks remain pronounced. These methods predominantly rely on predefined fixed-path scanning to flatten 2D images into 1D sequences. Yet, real-world cracks often manifest as winding and bifurcated structures with arbitrary orientations (Lang et al., 2024). This flattening process inherently disrupts the spatial contiguity of curved cracks, leading to feature fragmentation and discontinuous segmentation, particularly within complex backgrounds (Chen et al., 2022).

Receptance Weighted Key Value (RWKV) (Peng et al., 2023) is emerging as a potential next-generation foundation model following Transformers (Dosovitskiy et al., 2021) and Mamba. Vision-RWKV (VRWKV) (Duan et al., 2024) successfully adapted it for visual perception via a bidirectional WKV operator, introducing it to the visual domain and circumventing the limitations of causal modeling. However, directly applying the original VRWKV (Duan et al., 2024) to crack segmentation proves insufficient, as its inherent spatial interaction and static decay mechanisms constitute new bottlenecks. Specifically, the original RWKV models typically rely on the standard Q-Shift mechanism (Duan et al., 2024). While this rigid approach may be effective for regular objects, it fails to capture the versatile geometric deformations of cracks that exhibit extreme topological irregularities. More critically, the linear attention substitute module used in existing models indiscriminately treats all spatial tokens equally by default. In complex scenes, cracks often occupy only a distinct minority of pixels (Chen et al., 2024), while the surroundings are dominated by disturbances from irrelevant features such as texture noise, shadows, and pavement particles. The absence of a dynamic, content-aware selective mechanism (Fei et al., 2024; Choe et al., 2024; Hou & Yu, 2024) implies that the model can't suppress the cumulative impact of noise-related tokens or reinforce topology-centric signals during recursive propagation, thereby diminishing its long-range modeling capability.

To solve the above problems, we present SCRWKV, an ultra-compact network endowed with robust global linear modeling and structural perception capabilities. Specifically, we construct the Structure-Field Encoder (SFE) as the backbone to explicitly establish the global topological continuity of fractures. The workflow initiates with a standalone Adaptive Multi-scale Cascaded Modulator (AMCM) acting as a Structure-Field Initializer, which pre-modulates discrete tokens into an incipient field enriched with preliminary geometric correlations. To further refine these features, we introduce the Structure-Calibrated Insight Unit (SCIU) as the fundamental building block, which synergistically integrates three innovative mechanisms to achieve holistic modeling. Geometry-Guided Bidirectional Structure Transformation (GBST) supersedes fixed neighborhood offsets by simulating the bidirectional propagation of stress fields, thereby explicitly establishing long-range topological de-

pendencies across the entire image. The embedded AMCM mines multi-scale details via cascaded large kernels; as illustrated in Figure 1(b), it compensates for the local granularity limitations of linear attention, achieving superior segmentation accuracy compared to alternative enhancement modules. Dy-WKV, incorporating Dynamic Self-Calibrating Decay (DSCD), serves as a content-aware modulator. By dynamically generating adaptive decay factors, it selectively suppresses noise while amplifying structure-dominant signals, thus preserving crack integrity. The Cross-Scale Harmonic Fusion (CSHF) decoder employs a scale-signature binding mechanism to bridge semantic gaps, dynamically orchestrating multi-level features into a coherent harmonic state for high-precision segmentation. As demonstrated in Figure 1(c), a 4-layer SCIU configuration attains optimal performance, generating the sharpest segmentation maps while effectively shielding against interference and maintaining the model's extreme lightweight efficiency. In summary, the main contributions of this paper are as follows:

- We propose SCRWKV, an ultra-compact network with only 1.22M parameters that synergizes global linear modeling with structural awareness. It redefines the efficiency-performance trade-off, delivering SOTA precision with minimal parameter overhead in complex scenarios.
- We develop the SFE backbone featuring the SCIU block, where GBST and AMCM capture topological continuity and fine morphological cues, respectively, while DSCD performs adaptive noise filtering. Furthermore, a CSHF decoder is introduced to effectively bridge the semantic gap via dynamic multi-level fusion.
- Experimental results demonstrate that SCRWKV significantly outperforms SOTA methods across diverse benchmarks. This validates that extreme compactness does not compromise performance, ensuring robust applicability in real-world industrial environments.

## 2 Related Works

### 2.1 Crack Segmentation Networks

Automatic crack detection has evolved from early heuristics to data-driven deep learning. Early CNN-based methods, such as DeepCrack (Liu et al., 2019) and FPHBN (Yang et al., 2019), leveraged hierarchical convolutions for feature extraction. While CNNs excel at capturing high-frequency textures via local inductive biases, their limited receptive fields inherently constrain long-range dependency modeling, often resulting in fragmented segmentation for continuous cracks. Vision Transformers (ViT) (Dosovitskiy et al., 2021; Vaswani et al., 2017; Xia et al., 2024) mitigate this by establishing global context via self-attention. However, their quadratic complexity imposes prohibitive memory and latency costs.

Recently, Selective State Space Models (SSMs), notably Mamba (Gu et al., 2022), have emerged as compelling al-

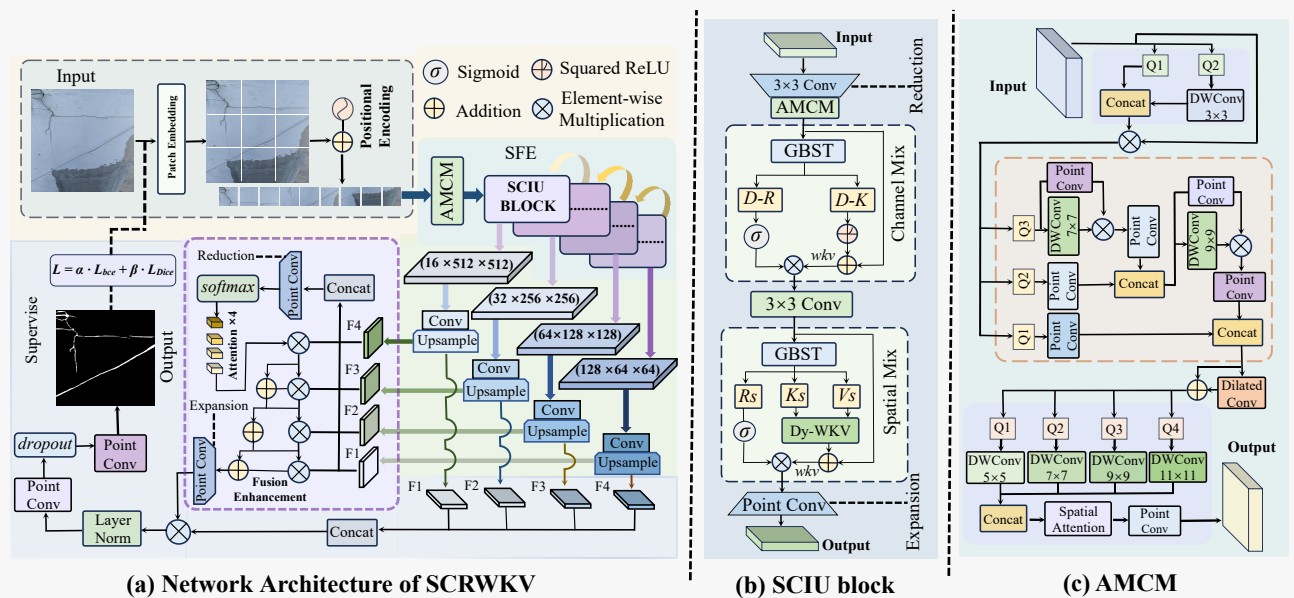

*Figure 2.* Overview of our proposed method. (a) Illustrates the overall architecture of SCRWKV and the processing flow for crack. (b) It displays the structure of the SCIU block, which integrates Spatial Mix and Channel Mix. It uses GBST for topological alignment and Dy-WKV to filter noise and model long-range dependencies. (c) Architecture of AMCM. It utilizes multi-scale large-kernel convolutions to efficiently expand the receptive field, while the adaptive gating mechanism ensures robust feature extraction in complex environments.

ternatives, offering global modeling with linear complexity. General visual backbones like ViM (Zhu et al., 2024), VMamba (Liu et al., 2024c), and PlainMamba (Yang et al., 2024) achieve Transformer-level receptive fields with CNN-like inference speeds. Specifically for crack segmentation, SCSegamba (Liu et al., 2025) employs a Structure-Aware Scan Strategy to enhance feature extraction. Nevertheless, Mamba-based methods (Liu et al., 2024b; Xing et al., 2024; Yang et al., 2024) face intrinsic limitations in modeling irregular topologies due to the rigid flattening of 2D images into 1D sequences. This serialization disrupts intrinsic spatial coherence, where spatially proximal pixels are widely separated in the 1D sequence, thereby impeding effective information propagation within the hidden state.

### 2.2 RWKV-based Visual Perception

Developing alongside SSMs, the Receptance Weighted Key Value (RWKV) (Peng et al., 2023) architecture synergizes Transformer-like parallel training with RNN-like inference efficiency. VRWKV (Duan et al., 2024) adapted this to vision tasks using bi-directional WKV operators for linear-complexity global modeling. However, applying existing RWKV paradigms to fine-grained crack segmentation remains suboptimal due to mechanical rigidities. Besides, the standard Q-Shift mechanism (Duan et al., 2024) enforces a deterministic, fixed-stride offset that inherently fails to trace the curvilinear manifolds and arbitrary bifurcations of structural cracks. Although Restore-RWKV (Yang et al., 2026) attempts to substitute spatial shifts with convolutions to capture fine-grained details, this approach compromises the core advantages of RWKV: its directional controllability

and linear simplicity. Convolutions introduce a heavy local bias that disrupts the integrity of global semantic modeling. Consequently, this design not only degrades the overall modeling performance but also imposes a severe computational burden, rendering it inefficient for high-resolution crack segmentation tasks.

Furthermore, standard linear attention mechanisms in these models employ a content-agnostic aggregation strategy. Lacking instance-based dynamic filtering (Li et al., 2025; Hou et al., 2025), they integrate high-frequency background noise into the hidden state, causing feature blurring in low signal-to-noise scenarios. To compensate for insufficient local priors, existing architectures often resort to brute-force depth stacking or dimension expansion (Duan et al., 2024), thereby incurring substantial parameter redundancy and computational overhead. Although existing methods have made significant progress, they remain suboptimal for deployment on resource-constrained edge devices.

## 3 Method

### 3.1 Preliminary

As illustrated in Figure 2(a), SCRWKV primarily comprises the SFE backbone for hierarchical feature extraction and the CSHF decoder for semantic aggregation. After Patch Embedding and Positional Encoding, the input is first processed by a standalone AMCM to initialize structural associations. Subsequently, the features are fed into the backbone composed of stacked SCIU blocks. Within each SCIU, the GBST explicitly constructs global topological continuity, the AMCM captures fine-grained local textures, and the

DSCD effectively suppresses background noise. The resulting multi-scale representations $\{F_1, \ldots, F_4\}$ are processed by the CSHF decoder, culminating in the generation of a precise binary segmentation mask.

## 3.2 Adaptive Multi-scale Cascaded Modulator

To accurately capture both microscopic hairline cracks and complex topological structures, we introduce the AMCM. This module employs a divide-and-conquer strategy, efficiently processing features through a hierarchical cascade followed by multi-scale refinement. Given input $\mathbf{X}_{in} \in \mathbb{R}^{C \times H \times W}$, we employ a parallel gating mechanism to enhance local details. Specifically, $\mathbf{X}_{in}$ is split into $\{\mathbf{X}_{p1}, \mathbf{X}_{p2}\}$, where $\mathbf{X}_{p2}$ undergoes a depthwise convolution before being concatenated with $\mathbf{X}_{p1}$ to modulate the features. This process is formulated as:

$$\tilde{\mathbf{X}} = \mathbf{X}_{in} \odot (\mathbf{X}_{p1} \oplus \mathcal{D}_3(\mathbf{X}_{p2})) \tag{1}$$

where $\mathcal{D}_k$ is a $k \times k$ depthwise convolution, $\oplus$ denotes channel-wise concatenation, and $\odot$ is the Hadamard product. We slice $\tilde{\mathbf{X}}$ into three subspaces $\mathcal{Q} = \{\mathbf{Q}_i\}_{i=1}^3$. To unify extraction across stages, we define a processing unit $\Psi_\kappa(\mathbf{h})$ parameterized by kernel scale $\kappa$:

$$\Psi_\kappa(\mathbf{h}) \triangleq \mathcal{W}_{p2}\left([(\mathcal{D}_\kappa \circ \delta \circ \mathcal{W}_{p1})(\mathbf{h})] \odot \mathcal{W}_{p1}(\mathbf{h})\right) \tag{2}$$

where $\mathcal{W}_{p \cdot}$ denote pointwise projections, $\delta$ is GELU, and $\circ$ is function composition. Subsequently, we formulate this context aggregation as a recursive state evolution process. With $\mathbf{Q}_2$ aligned via $\mathcal{W}_{proj}$, the state $\mathbf{H}_t$ updates as:

$$\mathbf{H}_t = \begin{cases} \Psi_7(\mathbf{Q}_3) & t = 1 \\ \Psi_9\left(\mathcal{W}_{proj}(\mathbf{Q}_2) \oplus \mathbf{H}_{t-1}\right) & t = 2 \end{cases} \tag{3}$$

The final aggregated feature $\mathbf{X}_{cas}$ is derived by fusing the terminal state with the basal subspace $\mathbf{Q}_1$, followed by a dilated convolution $\mathcal{R}_d$ to expand the receptive field:

$$\mathbf{X}_{cas} = \mathcal{R}_d\left(\mathbf{Q}_1 \oplus \mathbf{H}_2\right) \tag{4}$$

To capture diverse crack morphologies, $\mathbf{X}_{cas}$ is distributed into four parallel branches. We employ a set of multi-granular kernels $\mathcal{K} = \{5, 7, 9, 11\}$ to extract features, which are then fused into a unified representation $\mathbf{Z}_{ms}$:

$$\mathbf{Z}_{ms} = \bigoplus_{\kappa \in \mathcal{K}} \mathcal{D}_\kappa(\mathbf{X}_{cas}) \tag{5}$$

To model global dependencies, we first project the features onto a grid $\mathbf{X}_{grid}$ via adaptive downsampling $\mathcal{G}_\downarrow$:

$$\mathbf{X}_{grid} = \mathcal{G}_\downarrow(\mathbf{Z}_{ms}) \in \mathbb{R}^{C \times G \times G} \tag{6}$$

Based on this grid representation, we infer dynamic weights $\Omega$ and biases $\beta$ to modulate a learnable topology matrix $\mathbf{S}$. The spatial attention map $\mathcal{A}_{grid}$ is formulated as:

$$\mathcal{A}_{grid} = \mathcal{G}_\uparrow\left[\text{softmax}\left(\frac{\Omega \odot \mathbf{S}}{\tau}\right) \times (\Omega + \beta)\right] \tag{7}$$

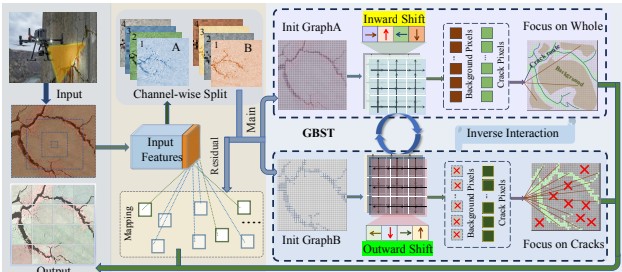

*Figure 3.* Schematic of the GBST. This mechanism explicitly models the curvilinear manifold of cracks by decoupling the feature space into two counter-directional streams.

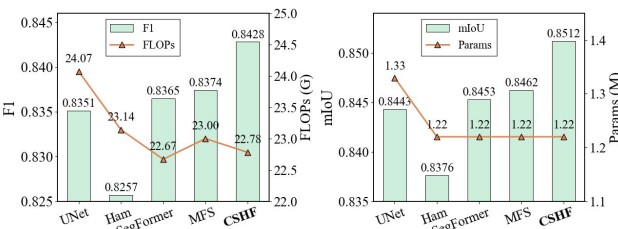

*Figure 4.* Comparison of different segmentation heads. The left and right subplots illustrate the scores for F1/FLOPs and mIoU/Params.

where $\mathcal{G}_\uparrow$ denotes bilinear upsampling, $\tau$ is a learnable temperature, and $\times$ represents the matrix multiplication for aggregating global context. The final spatial output is obtained via residual connection:

$$\mathbf{X}_{out} = \mathbf{X}_{in} + \mathcal{W}_{out}(\mathbf{Z}_{ms} \odot \sigma(\mathcal{A}_{grid})) \tag{8}$$

where $\mathcal{W}_{out}$ is output projection and $\sigma$ is Sigmoid.

## 3.3 Structure-Calibrated Insight Unit

Standard VRWKV architectures face limitations in structural defect segmentation due to rigid spatial priors and content-agnostic decay. To address these constraints, we propose the SCIU, as illustrated in Figure 2(b). It integrates the GBST, shown in Figure 3, to transcend fixed shifts. GBST partitions feature channels into counter-propagating streams where the Outward-shift Group $x_{out}$ diffuses signals to capture fracture width and the Inward-shift Group $x_{in}$ converges features to preserve the topological skeleton. Formally, let $\mathcal{T}_\Delta(\cdot)$ denote a spatial displacement operator that translates a feature map by a vector $\Delta \in \{\uparrow, \downarrow, \leftarrow, \rightarrow\}$. We define a directional shift mapping $\Phi$ that aggregates four-way spatial displacements:

$$\Phi(\mathbf{x}, \Delta) = \sum_{k=1}^4 \mathcal{T}\Delta_k(\mathbf{x}^{(k)}) \tag{9}$$

The complete GBST is then formulated as the concatenation of these counter-propagating streams:

$$\text{GBST}(\mathbf{X}_{in}) = \left[\Phi(\mathbf{x}_{out}, \Delta^{out}); \Phi(\mathbf{x}_{in}, \Delta^{in})\right] \tag{10}$$

where $[\cdot; \cdot]$ denotes the concatenation along the channel dimension, and $\Delta^{in} = -\Delta^{out}$ ensures bidirectional symmetry for structural rectification.

In the Spatial Mix phase, we first align input $\mathbf{X}_{in}$ via GBST and interpolate with a learnable gate $\mu_c$:

$$\tilde{\mathbf{x}}_c = \mu_c \odot \mathbf{X}_{in} + (1 - \mu_c) \odot \text{GBST}(\mathbf{X}_{in}) \quad (11)$$

From these aligned features, we derive the receptance $\mathbf{r}$, key $\mathbf{k}$, and value $\mathbf{v}$. Standard Vision-RWKV relies on a static spatial decay vector $w$. To capture local topological complexity, we propose Dy-WKV, introducing the DSCD defined as $\hat{\mathbf{W}}$:

$$\hat{\mathbf{W}} = w_{base} \odot \exp\left(-\sigma\left(\mathcal{W}_{decay}(\mathbf{X}_{in})\right)\right) \quad (12)$$

where $w_{base}$ is the learnable base decay and $\mathcal{W}_{decay}$ denotes the semantic projection. To capture the structural dependencies, we first introduce the structure-aware attention score $\mathcal{E}_{t,i}$ between tokens $t$ and $i$:

$$\mathcal{E}_{t,i} = \mathbf{k}_i - \frac{|t - i| - 1}{T} \odot \hat{\mathbf{W}} \quad (13)$$

The aggregated feature for the $t$-th token is then derived via the weighted bidirectional summation:

$$\text{Dy-WKV}_t = \frac{\sum_{i \neq t} e^{\mathcal{E}_{t,i}} \mathbf{v}_i + e^{\mathbf{u}+\mathbf{k}_t} \mathbf{v}_t}{\sum_{i \neq t} e^{\mathcal{E}_{t,i}} + e^{\mathbf{u}+\mathbf{k}_t}} \quad (14)$$

Here, the dynamic term $\hat{\mathbf{W}}$ spatially modulates the effective receptive field. The final spatial output is obtained via a residual connection:

$$\mathbf{Y}_{spatial} = \sigma(\mathbf{r}) \odot \text{Dy-WKV}_t + \mathbf{X}_{in} \quad (15)$$

In the Channel Mix phase, we employ a Dynamic Mixing Strategy. A dynamic mixing factor $\mathbf{D}_{dyn}$ is inferred via a context projection $\mathcal{W}_{ctx}$:

$$\mathbf{D}_{dyn} = \sigma\left(\mathcal{W}_{ctx}(\mathbf{X}_{in})\right) \quad (16)$$

The effective mixing weights $\mathbf{\Omega}_k$ are derived to modulate the GBST-transformed features:

$$\mathbf{x}'_k = \mathbf{\Omega}_k \odot \mathbf{X}_{in} + (\mathbf{1} - \mathbf{\Omega}_k) \odot \text{GBST}(\mathbf{X}_{in}) \quad (17)$$

The final output is activated via Squared-ReLU ($\phi$):

$$\mathbf{Y}_{channel} = \sigma(\mathcal{W}_r \mathbf{x}'_r) \odot \mathcal{W}_v \phi(\mathcal{W}_k \mathbf{x}'_k) + \mathbf{X}_{in} \quad (18)$$

where $\mathcal{W}_r, \mathcal{W}_k, \mathcal{W}_v$ are the learnable linear projections.

### 3.4 Cross-Scale Harmonic Fusion

To bridge the semantic discrepancy between high-level contexts and low-level boundaries, CSHF employs a scale-encoding and gated-fusion mechanism. We project backbone features $\{C_i\}_{i=1}^4$ to a unified channel dimension $D$ and align them to resolution $H \times W$ via content-aware DySample:

$$\tilde{\mathbf{F}}_i = \text{DySample}_{s_i}\left(\mathcal{W}p(C_i)\right) \quad (19)$$

where $\mathcal{W}_p$ is a pointwise projection and $s_i$ is the upsampling factor. To compensate for resampling loss, we inject a learnable scale embedding $\mathbf{E}_i \in \mathbb{R}^D$ into each feature, yielding

the topology-aware feature $\hat{\mathbf{F}}_i$:

$$\hat{\mathbf{F}}_i = \tilde{\mathbf{F}}_i + \mathbf{E}_i \quad (20)$$

where addition implies channel-wise broadcasting. Subsequently, we synthesize a harmonized representation through a scale-aware attention mechanism. We stack the features to generate a spatial attention map $\mathcal{A}_{scale}$:

$$\mathcal{A}_{scale} = \text{softmax}\left(\mathcal{W}_{attn}\left(\bigparallel_{i=1}^4 \hat{\mathbf{F}}_i\right)\right) \quad (21)$$

where $\parallel$ denotes channel-wise concatenation and $\mathcal{W}_{attn}$ serves as the spatial predictor. Guided by these weights, the harmonized feature $\mathbf{F}_{harm}$ is derived via:

$$\mathbf{F}_{harm} = \sum_{i=1}^4 \mathcal{A}_{scale_i} \odot \hat{\mathbf{F}}_i \quad (22)$$

Finally, we employ $\mathbf{F}_{harm}$ as a harmonic gate to modulate the full spectrum of original aligned features via a soft-masking mechanism. The final segmentation map $\mathbf{O} \in \mathbb{R}^{1 \times H \times W}$ is obtained by projecting the gated features into the output space:

$$\mathbf{O} = \mathcal{B}\left(\text{LN}\left(\mathcal{W}_{exp}(\mathbf{F}_{harm}) \odot \bigparallel_{i=1}^4 \tilde{\mathbf{F}}_i\right)\right) \quad (23)$$

where $\mathcal{W}_{exp}$ denotes channel expansion, LN represents Layer Normalization, $\mathcal{B}$ is the bottleneck convolution.

## 4 Experiments

### 4.1 Datasets

To evaluate the efficacy of the proposed method, comprehensive experiments are conducted on four public datasets: Crack500, DeepCrack, CrackMap, and TUT.

**Crack500** (Yang et al., 2019). Captured primarily via mobile devices, this dataset focuses on asphalt pavement scenarios. Through data augmentation techniques, the initial 500 images were expanded to 3,368 samples, each accompanied by precise pixel-level binary annotations.

**DeepCrack**(Liu et al., 2019). This dataset comprises 537 RGB images covering various material surfaces such as concrete, brick, and asphalt. It is characterized by diverse crack morphologies, ranging from fine continuous lines to wide, blurred boundaries. It also includes challenging conditions like oil stains, ensuring a comprehensive assessment of the model's representational capability.

**CrackMap** (Katsamenis et al., 2023). Designed explicitly for high-precision road inspection, this dataset consists of 120 high-resolution RGB images. Targeting subtle and topologically complex asphalt cracks challenges the model's ability to resolve intricate high-frequency details.

**TUT** (Liu et al., 2024a). Distinguished from traditional datasets by its dense and cluttered backgrounds, this dataset presents the most complex interference scenarios. It con-

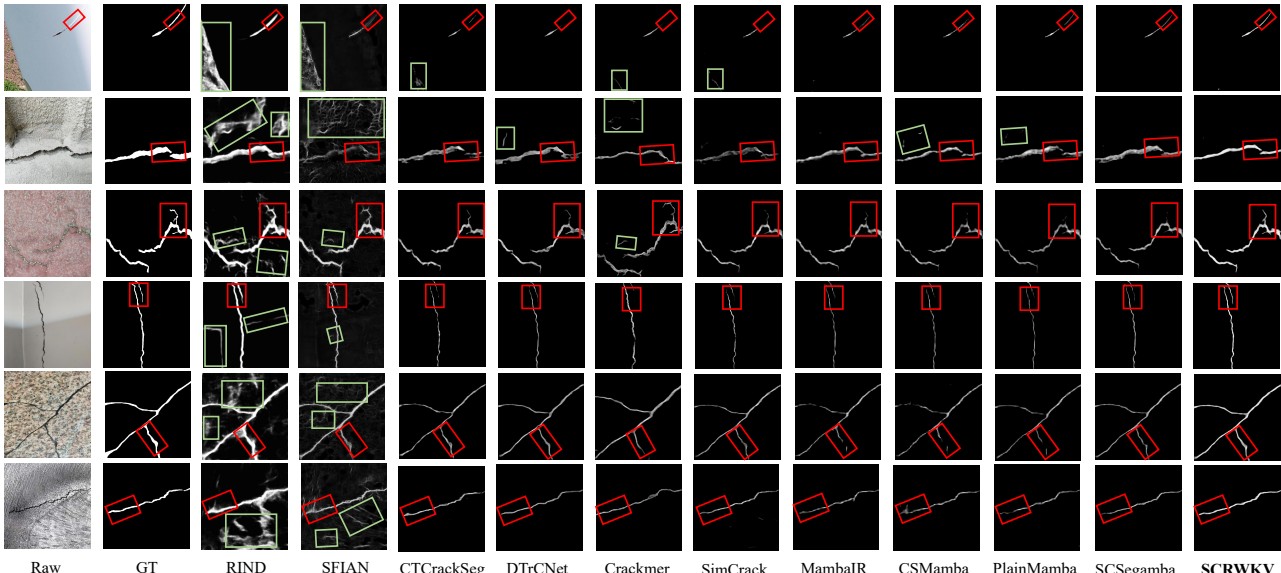

*Figure 5.* Visual comparison of typical crack segmentation results on the TUT dataset against 10 SOTA methods. Red boxes highlight critical details, and green boxes mark misidentified regions.

| Methods | Crack500 | | | | | | DeepCrack | | | | | |
|---|---|---|---|---|---|---|---|---|---|---|---|---|
| | ODS | OIS | P | R | F1 | mIoU | ODS | OIS | P | R | F1 | mIoU |
| RIND | 0.6469 | 0.6483 | 0.6998 | 0.7245 | 0.7119 | 0.7381 | 0.8087 | 0.8267 | 0.7896 | 0.8920 | 0.8377 | 0.8391 |
| SFIAN | 0.6977 | 0.7348 | 0.6983 | 0.7742 | 0.7343 | 0.7604 | 0.8616 | 0.8928 | 0.8549 | 0.8692 | 0.8620 | 0.8776 |
| CTCrackSeg | 0.6941 | 0.7059 | 0.6940 | 0.7748 | 0.7322 | 0.7591 | 0.8819 | 0.8904 | 0.9011 | 0.8895 | 0.8952 | 0.8925 |
| DTrCNet | 0.7012 | 0.7241 | 0.6527 | **0.8280** | 0.7357 | 0.7627 | 0.8473 | 0.8512 | 0.8905 | 0.8251 | 0.8566 | 0.8661 |
| Crackmer | 0.6933 | 0.7097 | 0.6985 | 0.7572 | 0.7267 | 0.7591 | 0.8712 | 0.8785 | 0.8946 | 0.8783 | 0.8864 | 0.8844 |
| SimCrack | 0.7127 | 0.7308 | 0.7093 | 0.7984 | 0.7516 | 0.7715 | 0.8570 | 0.8722 | 0.8984 | 0.8549 | 0.8761 | 0.8744 |
| MambaIR | 0.7043 | 0.7189 | 0.7204 | 0.7681 | 0.7435 | 0.7663 | 0.8796 | 0.8840 | 0.9056 | 0.8895 | 0.8975 | 0.8907 |
| CSMamba | 0.6931 | 0.7162 | 0.6858 | 0.7823 | 0.7315 | 0.7592 | 0.8738 | 0.8766 | 0.9025 | 0.8863 | 0.8943 | 0.8863 |
| PlainMamba | 0.7035 | 0.7173 | 0.7170 | 0.7557 | 0.7358 | 0.7682 | 0.8646 | 0.8668 | 0.9050 | 0.8659 | 0.8850 | 0.8788 |
| SCSegamba | 0.7244 | 0.7370 | **0.7270** | 0.7859 | **0.7553** | 0.7778 | 0.8938 | 0.8990 | 0.9097 | 0.9124 | 0.9110 | 0.9022 |
| **SCRWKV (Ours)** | **0.7256** | **0.7447** | 0.7013 | 0.8180 | 0.7552 | **0.7787** | **0.9254** | **0.9307** | **0.9211** | **0.9520** | **0.9363** | **0.9289** |
| Methods | CrackMap | | | | | | TUT | | | | | |
| | ODS | OIS | P | R | F1 | mIoU | ODS | OIS | P | R | F1 | mIoU |
| RIND | 0.6745 | 0.6943 | 0.6023 | 0.7586 | 0.6699 | 0.7425 | 0.7531 | 0.7891 | 0.7872 | 0.7665 | 0.7767 | 0.8051 |
| SFIAN | 0.7200 | 0.7465 | 0.6715 | 0.7668 | 0.7160 | 0.7748 | 0.7290 | 0.7513 | 0.7715 | 0.7367 | 0.7537 | 0.7896 |
| CTCrackSeg | 0.7289 | 0.7373 | 0.6911 | 0.7669 | 0.7270 | 0.7785 | 0.7940 | 0.7996 | 0.8202 | 0.8195 | 0.8199 | 0.8301 |
| DTrCNet | 0.7328 | 0.7413 | 0.6912 | 0.7681 | 0.7276 | 0.7812 | 0.7987 | 0.8073 | 0.7972 | 0.8441 | 0.8202 | 0.8331 |
| Crackmer | 0.7395 | 0.7437 | 0.7229 | 0.7467 | 0.7346 | 0.7860 | 0.7429 | 0.7640 | 0.7501 | 0.7656 | 0.7578 | 0.7966 |
| SimCrack | 0.7559 | 0.7625 | 0.7380 | 0.7672 | 0.7523 | 0.7963 | 0.7984 | 0.8090 | 0.8051 | 0.8371 | 0.8208 | 0.8334 |
| MambaIR | 0.7332 | 0.7347 | 0.7569 | 0.7013 | 0.7280 | 0.7834 | 0.7861 | 0.7930 | 0.7877 | 0.8387 | 0.8125 | 0.8249 |
| CSMamba | 0.7371 | 0.7413 | 0.7053 | 0.7663 | 0.7346 | 0.7841 | 0.7879 | 0.7946 | 0.7947 | 0.8353 | 0.8146 | 0.8263 |
| PlainMamba | 0.7150 | 0.7189 | 0.6649 | 0.7616 | 0.7099 | 0.7699 | 0.7867 | 0.7967 | 0.7701 | 0.8523 | 0.8102 | 0.8253 |
| SCSegamba | 0.7741 | 0.7766 | 0.7629 | 0.7727 | 0.7678 | 0.8094 | 0.8204 | 0.8255 | 0.8241 | 0.8545 | 0.8390 | 0.8479 |
| **SCRWKV (Ours)** | **0.7756** | **0.7811** | **0.7655** | **0.7826** | **0.7740** | **0.8106** | **0.8245** | **0.8313** | **0.8213** | **0.8655** | **0.8428** | **0.8512** |

*Table 1.* Comparison with 10 SOTA methods across 4 datasets. Best results are in green , and second-best results are blue .

tains 1,408 RGB images spanning eight distinct structural backgrounds, including asphalt, concrete, runways, tiles, and metal blades, featuring highly intricate and complex crack shapes.

### 4.2 Implementation Details

**Experimental Settings.** The SCRWKV framework was implemented using PyTorch v1.13.1. All experiments were conducted on a high-performance server equipped with an Intel Xeon Platinum 8336C CPU and eight NVIDIA GeForce RTX 4090 GPUs. The model was optimized using the AdamW optimizer with an initial learning rate of $5 \times 10^{-4}$, adjusted via a Poly learning rate (PolyLR) scheduling strategy. The weight decay was set to 0.01, and the random seed was fixed at 42 to ensure reproducibility. The network was trained for 50 epochs, and the checkpoint

| Methods | Year | FLOPs↓ | Param.↓ | Size↓ |
|---|---|---|---|---|
| RIND | ICCV 2021 | 695.77G | 59.39M | 453MB |
| SFIAN | TITS 2023 | 84.57G | 13.63M | 56MB |
| CTCrackSeg | ICIP 2023 | 39.47G | 22.88M | 174MB |
| DTrCNet | AIC 2023 | 123.20G | 63.45M | 317MB |
| Crackmer | AIC 2024 | **14.94G** | 5.90M | 43MB |
| SimCrack | WACV 2024 | 286.62G | 29.58M | 225MB |
| MambaIR | ECCV 2024 | 47.32G | 10.34M | 79MB |
| CSMamba | arXiv 2024 | 145.84G | 35.95M | 233MB |
| PlainMamba | BMVC 2024 | 73.36G | 16.72M | 96MB |
| SCSegamba | CVPR 2025 | 18.16G | 2.80M | 37MB |
| **SCRWKV** | ICML 2026 | 22.78G | **1.22M** | **28MB** |

*Table 2.* Comparison of complexity with other methods. Best results are in green, and second-best results are blue.

achieving the best performance on the validation set was selected for final testing.

**Comparison Methods.** To comprehensively evaluate the effectiveness of the proposed model, we benchmarked SCR-WKV against 10 SOTA methods. These baselines are categorized into two groups: CNN- or Transformer-based models, including RIND (Pu et al., 2021), SFIAN (Cheng et al., 2023), CTCrackSeg (Tao et al., 2023), DTrCNet (Xiang et al., 2023), Crackmer (Wang et al., 2024) and SimCrack (Jaziri et al., 2024); Representative Mamba-based models, such as CSMamba (Liu et al., 2024b), PlainMamba (Yang et al., 2024), MambaIR (Guo et al., 2024), and SC-Segamba(Liu et al., 2025).

**Evaluation Metrics.** To thoroughly evaluate the performance of SCRWKV, we employ six core metrics: Precision (P), Recall (R), F1 Score ($F1 = \frac{2RP}{R+P}$), Optimal Dataset Scale (ODS), Optimal Image Scale (OIS), and Mean Intersection over Union (mIoU). Specifically, ODS measures the model's adaptability to the entire dataset under a global fixed threshold $m$, whereas OIS evaluates the adaptability under an optimal threshold $n$ selected for each individual image. The calculation formulas are as follows:

$$ODS = \max_{m} \frac{2 \cdot P_m \cdot R_m}{P_m + R_m} \tag{24}$$

$$OIS = \frac{1}{N_{img}} \sum_{i=1}^{N_{img}} \max_{n} \frac{2 \cdot P_{n,i} \cdot R_{n,i}}{P_{n,i} + R_{n,i}} \tag{25}$$

mIoU is utilized to measure the average intersection-over-union ratio between ground truth labels and predicted results, serving as a pivotal metric for segmentation accuracy. The formula is defined as:

$$mIoU = \frac{1}{N+1} \sum_{l=0}^{N} \frac{P_{ll}}{\sum_{t=0}^{N} P_{tl} + \sum_{t=0}^{N} P_{lt} - P_{ll}} \tag{26}$$

where $N$ denotes the number of classes; $t$ represents the

ground truth label, $l$ represents the predicted value, and $P_{tl}$ indicates the number of pixels belonging to class $t$ but classified as $l$. Furthermore, three metrics are employed to assess the complexity of the method: FLOPs, Params, and Model Size, representing computational complexity, parametric scale, and memory footprint, respectively.

### 4.3 Comparison with SOTA Methods

As shown in Table 1, compared with 10 other SOTA methods including MambaIR(Guo et al., 2024), CSMamba (Liu et al., 2024b), and SCSegamba (Liu et al., 2025), the proposed SCRWKV achieves superior performance across all four public datasets. Specifically, on the DeepCrack (Liu et al., 2019) and Crack500 (Yang et al., 2019) datasets, which contain clear yet topologically complex crack regions, SCRWKV demonstrates a significant performance lead. Notably, on the DeepCrack (Liu et al., 2019) dataset, SCRWKV achieves an F1 score of 0.9363 and an mIoU of 0.9289, outperforming the second-best method by margins of 2.53 % and 2.67%, respectively. This substantial improvement is primarily attributable to the AMCM, which drastically enhances the model's capability to capture crack morphological features and fine boundary details. On the CrackMap (Katsamenis et al., 2023) dataset, characterized by subtler and longer continuous cracks, our method surpasses all other SOTA methods across all metrics, achieving an F1 score of 0.7740 and an mIoU of 0.8106. This serves as evidence for the effectiveness of the GBST in establishing bidirectional topological correlations and capturing fine textures and crack structures, circumventing fragmentation issues encountered in long-range modeling.

For the TUT dataset (Liu et al., 2024a), which encompasses eight distinct complex scenarios, our method similarly achieves SOTA performance, with F1 and mIoU reaching 0.8428 and 0.8512, respectively. As illustrated in Figure 5, SCRWKV consistently generates high-quality segmentation maps, whether on metal surfaces with high-noise backgrounds or in dimly lit environments with low contrast. This indicates that the DSCD operator integrated into the Dy-WKV plays a pivotal role by dynamically suppressing irrelevant environmental noise based on channel-wise responses. Furthermore, feature aggregation via CSHF enhances multi-scale perception, enabling our model to robustly handle diverse, interference-rich real-world engineering scenarios despite having an extremely lightweight parameter count of only 1.22M.

### 4.4 Complexity Analysis

Table 2 presents a detailed complexity comparison between our method and other SOTA approaches, with the input image resolution uniformly set to $512 \times 512$. SCRWKV demonstrates superior lightweight characteristics, requiring only 1.22M parameters and a model size of 28MB, thereby outperforming all comparative methods in terms of parame-

| AMCM | GBST | DSCD | F1 | mIoU | Param.↓ | FLOPs↓ |
|---|---|---|---|---|---|---|
| ✓ | × | × | 0.8359 | 0.8461 | 1.21M | 22.51G |
| × | ✓ | × | 0.8346 | 0.8450 | **0.76M** | **16.59G** |
| × | × | ✓ | 0.8340 | 0.8426 | 0.77M | 16.87G |
| ✓ | ✓ | × | 0.8340 | 0.8460 | 1.21M | 22.51G |
| ✓ | × | ✓ | 0.8381 | 0.8470 | 1.22M | 22.78G |
| × | ✓ | ✓ | 0.8295 | 0.8392 | 0.77M | 16.87G |
| ✓ | ✓ | ✓ | **0.8428** | **0.8512** | 1.22M | 22.78G |

*Table 3.* Ablation of components within SCRWKV. Best results are in green , and second-best results are blue .

| Head | ODS | OIS | R | F1 | mIoU |
|---|---|---|---|---|---|
| UNet | 0.8142 | 0.8193 | 0.8401 | 0.8351 | 0.8443 |
| Ham | 0.8037 | 0.8113 | 0.8247 | 0.8257 | 0.8376 |
| SegFormer | 0.8164 | 0.8235 | 0.8496 | 0.8365 | 0.8453 |
| MFS | 0.8087 | 0.8250 | 0.8602 | 0.8374 | 0.8462 |
| **CSHF (Ours)** | **0.8245** | **0.8313** | **0.8655** | **0.8428** | **0.8512** |

*Table 4.* Ablation study of different segmentation heads. Best results are in green , and second-best results are blue .

| Methods | ODS | OIS | P | F1 | mIoU |
|---|---|---|---|---|---|
| bi_ParaSnake | 0.8149 | 0.8208 | 0.8249 | 0.8353 | 0.8452 |
| DiagSnake | 0.8187 | 0.8267 | 0.8153 | 0.8397 | 0.8476 |
| bi_DiagSnake | 0.8182 | 0.8268 | 0.8099 | 0.8369 | 0.8465 |
| SASS | 0.7951 | 0.8022 | 0.8221 | 0.8141 | 0.8310 |
| Omishift | 0.8185 | 0.8256 | 0.8067 | 0.8388 | 0.8478 |
| Q-Shift | 0.8183 | 0.8229 | **0.8327** | 0.8381 | 0.8470 |
| **GBST (Ours)** | **0.8245** | **0.8313** | 0.8213 | **0.8428** | **0.8512** |

*Table 5.* Comparison of interaction mechanisms. Best results are in green , and second-best results are blue .

ter efficiency. Specifically, it achieves reductions of 56.43% and 22.22% in parameter count and model size, respectively, compared to the second-best result. Although SCRWKV exhibits a marginal increase in FLOPs compared to the computation-efficient Crackmer (Wang et al., 2024), this slight computational cost is effectively traded for a substantial reduction in parameter count and a significant boost in segmentation accuracy. This confirms that the synergy of SFE and CSHF ensures robust segmentation in noisy scenarios with minimal storage, making it ideal for deployment on resource-constrained edge devices like UAVs.

### 4.5 Ablation Studies

**Ablation Study on Core Components.** As shown in Table 3, we conducted a stepwise decoupling analysis to quantify the impact of AMCM, GBST, and DSCD. The AMCM backbone serves as the foundational feature extractor; while increasing parameters from 0.77M to 1.22M, it breaks the representation bottleneck of purely lightweight configurations, yielding a significant 1.20% mIoU leap. This confirms that structural perception cannot be sacrificed for extreme compactness. Furthermore, GBST and DSCD demonstrate exceptional parameter efficiency. Compared to the AMCM-

only baseline, integrating these mechanisms incurs negligible cost yet boosts mIoU and F1 by 0.51% and 0.69%, respectively. The suboptimal performance of independent variants indicates that GBST and DSCD provide complementary optimization in spatial and channel domains, effectively resolving topological disconnections in fine cracks.

**Ablation Study on Segmentation Heads.** As illustrated in Table 4 and Figure 4, the integration of our designed CSHF segmentation head enables the model to achieve optimal performance across key metrics. Specifically, the F1 score and mIoU reach 0.8428 and 0.8512, respectively, validating its effectiveness in modeling fine-grained crack structures. Notably, CSHF maintains an identical lightweight footprint as the MFS (Liu et al., 2025), SegFormer (Xie et al., 2021), and Ham heads (Geng et al., 2021), which are uniformly configured with exactly 1.22M parameters. Under these equivalent parameter constraints, CSHF outperforms the second-best MFS head (Liu et al., 2025). While operating with slightly lower computational overhead than MFS, it elevates the F1 score from 0.8374 to 0.8428 and boosts mIoU from 0.8462 to 0.8512. This confirms that CSHF's cross-scale fusion strategy maximizes feature representation efficiency without incurring additional parameter costs.

**Ablation on Spatial Structural Interaction Mechanisms.** As presented in Table 5, under identical training configurations, distinct spatial interaction mechanisms exert a significant influence on model performance. Compared to conventional spatial interaction mechanisms such as DiagSnake, Q-Shift (Duan et al., 2024), and SASS (Liu et al., 2025), our proposed GBST achieves SOTA performance across most metrics. Specifically, relative to the closest competing methods, GBST yields consistent improvements, outperforming the second-best DiagSnake by 0.31% in F1 score and surpassing Omishift by 0.34% in mIoU . This confirms that unlike rigid scanning paths, GBST's dynamic geometric field enables more effective feature interaction, precisely capturing the intricate topology of cracks.

## 5 Conclusion

This paper presents SCRWKV, a framework that successfully overcomes the bottleneck of balancing topological modeling capability and computational efficiency in crack segmentation. SCRWKV combines SFE and CSHF to enhance crack shape and texture perception. Utilizing the AMCM and the SCIU which integrates GBST and DSCD, the SFE synergistically resolves long-range dependency fragmentation and environmental noise interference, achieving precise capture of fine textures and non-linear geometric structures. Experiments demonstrate that SCRWKV attains SOTA performance, achieving an F1 score of 0.8428 and an mIoU of 0.8512, with an extremely lightweight footprint of only 1.22M parameters. This optimal trade-off establishes SCRWKV as a premier solution for Structural Health Moni-

toring on resource-constrained edge devices. Future work will focus on generalizing this topology-aware paradigm to fine-grained segmentation tasks, while further investigating its scalability in high-resolution, complex scenarios.

## Impact Statement

This paper presents work aimed at advancing efficient crack segmentation for structural health monitoring, facilitating low-cost and automated inspections of civil infrastructure. There are many potential societal consequences of our work, none of which we feel must be specifically highlighted here.

## Acknowledgement

This work was supported by the National Natural Science Foundation of China (NSFC) under Grants T2422015 and 62306212; the Beijing-Tianjin-Hebei Natural Science Foundation Cooperation Project under Grant 25JJJJC0009; the China Postdoctoral Science Foundation under Grant 2024M762376; and the Marie Sklodowska-Curie Actions (MSCA) under Project No. 10111188.

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

# SCRWKV: Ultra-Compact Structure-Calibrated Vision-RWKV for Topological Crack Segmentation

## Supplementary Material

## 6 Details of Objective Function and Analysis

To achieve an optimal equilibrium between pixel-level classification accuracy and holistic structural overlap supervision, we employ a hybrid loss function $L_{total} = \alpha L_{BCE} + \beta L_{Dice}$ to jointly optimize the model. The calculation formulas for $L_{BCE}$ (Li et al., 2024) and $L_{Dice}$ (Sudre et al., 2017) are defined as follows:

$$L_{Dice} = 1 - \frac{2 \sum_{j=1}^{M} p_j \hat{p}_j + \epsilon}{\sum_{j=1}^{M} p_j + \sum_{j=1}^{M} \hat{p}_j + \epsilon} \quad (27)$$

$$L_{BCE} = -\frac{1}{N} \left[ p_j \log(\hat{p}_j) + (1 - p_j) \log(1 - \hat{p}_j) \right] \quad (28)$$

To determine the optimal configuration for the loss function hyperparameters $\alpha$ ( BCE weight) and $\beta$ (Dice weight), we conducted a systematic sensitivity analysis to explore their impact on crack detection accuracy. As presented in Table 6, we evaluated the model performance across a broad range of Dice:BCE ratios, spanning from 1:5 to 5:1. To establish a robust baseline, these composite configurations were compared against singular loss strategies where the model was trained exclusively with either binary cross-entropy or Dice loss in isolation. The experimental results indicate that the model achieves the optimal performance trade-off when the ratio is set to Dice:BCE = 3:1, yielding a peak F1 score of 0.8428 and an mIoU of 0.8512. This specific allocation of weights effectively balances the local and global supervision signals. By moderately emphasizing structural consistency through the Dice loss component, the model is better equipped to overcome the extreme class imbalance inherent in crack segmentation, where crack pixels occupy only a small fraction of the total image area. Retaining fundamental pixel-level supervision via the BCE component is essential for maintaining sharp boundary fidelity and preventing the over-smoothing of fine crack details.

| $\beta:\alpha$ | ODS | OIS | P | R | F1 | mIoU |
|---|---|---|---|---|---|---|
| 1:5 | 0.8218 | 0.8287 | 0.8218 | 0.8613 | 0.8411 | 0.8494 |
| 1:4 | 0.8208 | 0.8251 | **0.8387** | 0.8446 | 0.8417 | 0.8488 |
| 1:3 | 0.8202 | 0.8251 | 0.8346 | 0.8455 | 0.8400 | 0.8483 |
| 1:2 | 0.8227 | 0.8288 | 0.8289 | 0.8540 | 0.8413 | 0.8497 |
| 1:1 | 0.8201 | 0.8277 | 0.8075 | 0.8695 | 0.8373 | 0.8483 |
| 2:1 | 0.8215 | 0.8298 | 0.8094 | **0.8722** | 0.8396 | 0.8489 |
| 3:1 | **0.8245** | **0.8313** | 0.8213 | 0.8655 | **0.8428** | **0.8512** |
| 4:1 | 0.8218 | 0.8280 | 0.8240 | 0.8579 | 0.8406 | 0.8492 |
| 5:1 | 0.8203 | 0.8252 | 0.8336 | 0.8477 | 0.8406 | 0.8482 |
| BCE | 0.8119 | 0.8184 | 0.8205 | 0.8512 | 0.8356 | 0.8432 |
| Dice | 0.8214 | 0.8274 | 0.8200 | 0.8599 | 0.8395 | 0.8484 |

*Table 6.* Sensitivity analysis experiments with different $\alpha$ and $\beta$ ratios. Best results are in green , and second-best results are blue .

et al., 2023), SCRWKV demonstrates exceptional performance in delineating microscopic, subtle cracks and preserving topological continuity. It effectively mitigates common issues found in CNN-based methods exemplified by RIND (Pu et al., 2021) and SFIAN (Cheng et al., 2023), such as boundary blurring and region dilation, while simultaneously avoiding the topological fragmentation often observed in Transformer or Mamba variants. This precision is explicitly attributed to the synergistic calibration of the AMCM and GBST, which capture multi-scale details while preserving the geometric integrity of the crack skeleton. Furthermore, on the TUT dataset (Liu et al., 2024a), which contains severe interference such as oil stains and complex pavement textures, SCRWKV exhibits remarkable immunity to false positives. Unlike competing models that frequently misclassify background artifacts as cracks, our framework leverages the DSCD operator within the SCIU to actively attenuate the weights of noise-dominant channels, thereby achieving precise segmentation even in low-contrast and cluttered scenarios.

## 7 Visualisation Comparisons

As illustrated in Figure 6, our qualitative comparisons across four benchmark datasets highlight the superiority of SCR-WKV in balancing morphological fidelity with environmental robustness. On datasets characterized by diverse crack widths and textures, such as Crack500 (Yang et al., 2019), DeepCrack (Liu et al., 2019), and CrackMap (Katsamenis

## 8 Additional Analysis

Due to space constraints in the main manuscript, we present further quantitative analyses in this supplementary material to thoroughly validate the necessity of each constituent module and substantiate the architectural design choices of SCRWKV. In this section, we provide a comprehensive evaluation. These extensive experiments collectively demon-

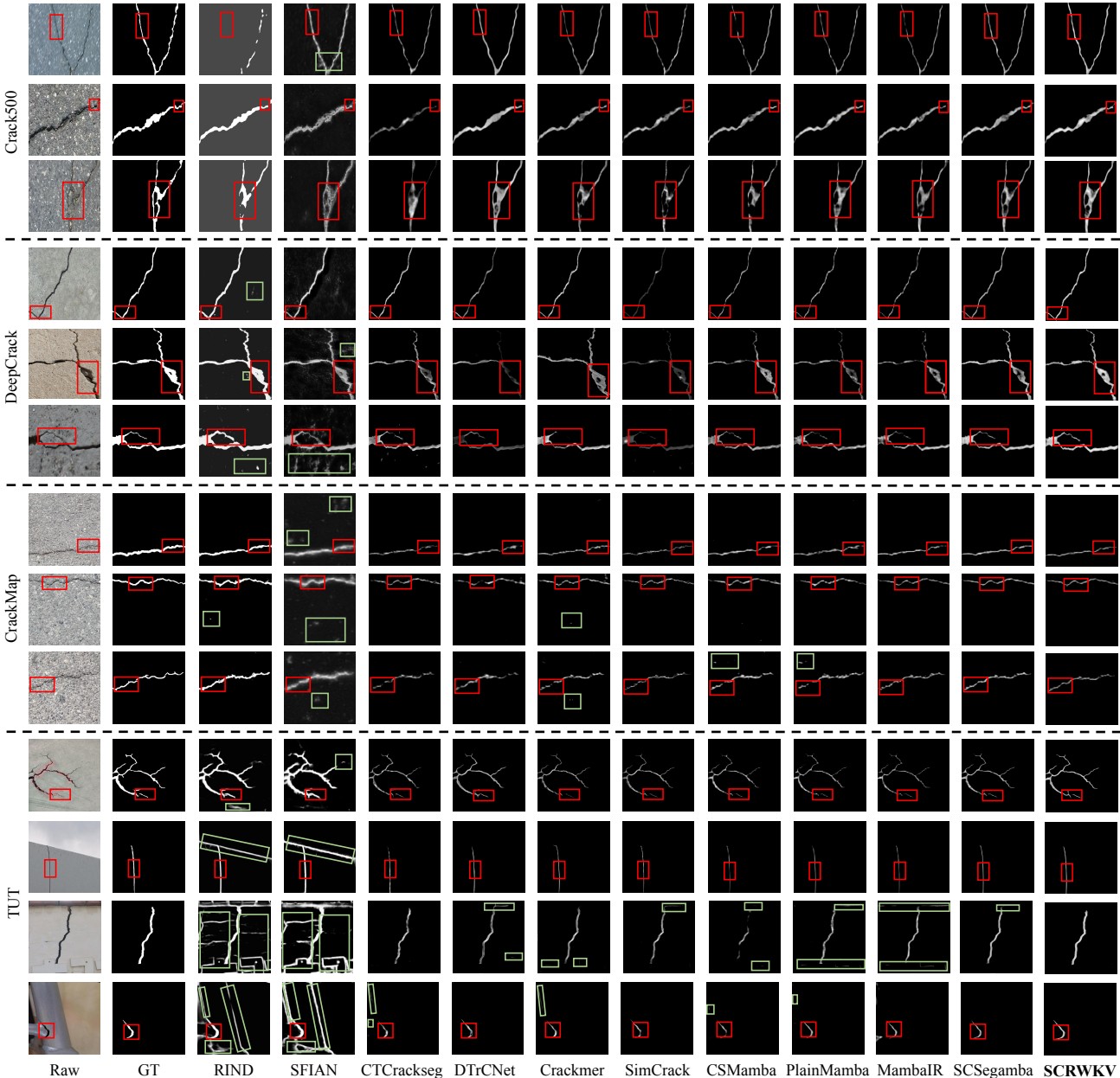

*Figure 6.* Visual comparison with 10 SOTA methods across four public datasets. Red boxes highlight critical details, and green boxes mark misidentified regions.

| Layer Num | ODS | OIS | P | R | F1 | mIoU | Params↓ | FLOPs↓ | Model Size↓ |
|---|---|---|---|---|---|---|---|---|---|
| 2 | 0.8117 | 0.8176 | **0.8302** | 0.8327 | 0.8314 | 0.8424 | **0.87M** | **17.03G** | **23MB** |
| 4 | **0.8245** | **0.8313** | 0.8213 | 0.8655 | **0.8428** | **0.8512** | 1.22M | 22.78G | 28MB |
| 8 | 0.8216 | 0.8289 | 0.8016 | **0.8791** | 0.8386 | 0.8489 | 1.91M | 34.78G | 39MB |
| 16 | 0.8219 | 0.8279 | 0.8224 | 0.8615 | 0.8415 | 0.8494 | 3.30M | 58.98G | 62MB |

*Table 7.* Comparison of different layer numbers. Best results are highlighted in green and the second best are blue .

| Patch Size | ODS | OIS | P | R | F1 | mIoU | Params↓ | FLOPs↓ | Model Size↓ |
|---|---|---|---|---|---|---|---|---|---|
| 4 | **0.8245** | **0.8313** | **0.8213** | 0.8655 | **0.8428** | **0.8512** | **1.22M** | 22.78G | 28MB |
| 8 | 0.8161 | 0.8232 | 0.8026 | **0.8685** | 0.8342 | 0.8448 | 1.42M | 10.77G | **24MB** |
| 16 | 0.7852 | 0.7909 | 0.7975 | 0.8138 | 0.8055 | 0.8232 | 2.21M | 7.77G | 32MB |
| 32 | 0.7161 | 0.7216 | 0.7284 | 0.7498 | 0.7390 | 0.7785 | 5.35M | **7.02G** | 69MB |

*Table 8.* Comparison of different patch sizes. Best results are highlighted in green and the second best are blue.

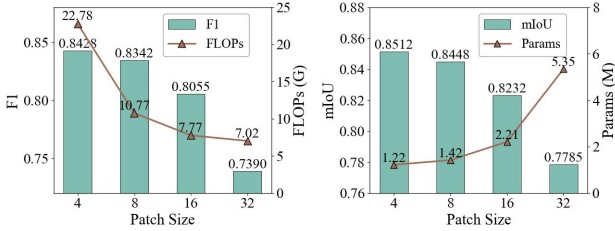

*Figure 7.* Comparison of different patch sizes. The left and right subplots illustrate the scores for F1/FLOPs and mIoU/Params.

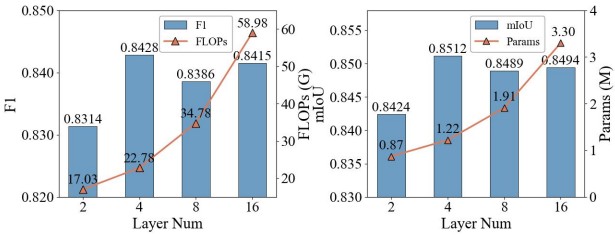

*Figure 8.* Comparison of different number of layers. The left and right subplots illustrate the scores for F1/FLOPs and mIoU/Params.

strate the robustness and superiority of the proposed method across diverse settings.

**Impact of SCIU Stacking Depth.** To achieve an optimal trade-off between segmentation accuracy and computational efficiency, we investigated the impact of stacking varying numbers of SCIUs within the encoder, evaluating configurations with $N \in \{2, 4, 8, 16\}$ layers. As illustrated in Table 7 and Figure 8 , the 4-layer configuration yielded the optimal results. Although the shallower 2-layer model exhibited the lowest resource consumption with only 0.87M parameters, its insufficient capacity for complex topological modeling limited the mIoU to just 0.8424. Conversely, increasing the depth to 8 or 16 layers did not yield performance gains; instead, it resulted in diminishing returns and feature redundancy. For instance, despite the 16-layer model nearly tripling the parameter count to 3.30M and reaching 58.98G FLOPs, its mIoU slightly decreased to 0.8494 compared to the 4-layer baseline. Consequently, we adopted the 4-layer structure as the default configuration. This setup secures a peak mIoU of 0.8512 and an F1 score of 0.8428 while utilizing only 1.22M parameters, making it highly suitable for practical deployment.

**Comparison of Different Patch Sizes.** During the Patch Embedding stage, the patch size $P$ dictates the granularity of the initial structural field. To identify the optimal resolution for crack feature extraction, we conducted experiments with $P \in \{4, 8, 16, 32\}$. As presented in Table 8, setting the patch size to 4 yielded the superior performance, achieving an F1 score of 0.8428 and an mIoU of 0.8512. In contrast, increasing the patch size resulted in a progressive degradation of segmentation quality. As illustrated in Figure 7, both F1 and mIoU scores exhibited a significant downward trend as the patch size increased. This is primarily attributed to the fact that structural cracks are inherently fine-grained targets; larger patches fail to preserve these high-frequency spatial details. This leads to the mixed pixel problem, where crack features become overwhelmed by the background information. Although larger patches reduce computational cost, they paradoxically increase the model size due to the quadratic growth of parameters in the embedding projection layer. The parameter count escalated from 1.22M at $P = 4$ to 5.35M at $P = 32$. Consequently, a patch size of 4 strikes the optimal balance, offering the highest accuracy while maintaining the most compact parameter count.

**Comparison under Identical Core Block Depth.** In Subsection 4.3, we compared our approach against other SOTA methods, where Mamba-based models, such as MambaIR (Guo et al., 2024), CSMamba (Liu et al., 2024b), PlainMamba (Yang et al., 2024), and SCSegamba (Liu et al., 2025), utilized their default network depths. To investigate the complexity and performance under a unified model depth and ensure a fair comparison of their underlying sequence modeling mechanisms, we standardized the number of core global modeling blocks to exactly 4 layers. Specifically, these core blocks refer to the VSS blocks in Mamba variants and the SCIU blocks in SCRWKV. All methods were evaluated under identical training configurations. The comparative results are presented in Table 2 and Table 9.

Experimental results indicate that when the number of core blocks is restricted to four, existing Mamba-based models exhibit varying degrees of degradation in segmentation performance, despite a reduction in computational complexity. For instance, under this 4-layer setting, CSMamba's F1 score and mIoU dropped to 0.7503 and 0.7773, respectively, demonstrating a strong dependency on deep state-space modeling. Although PlainMamba (Yang et al., 2024)

| Methods | ODS | OIS | P | R | F1 | mIoU | Params↓ | FLOPs↓ | Model Size↓ |
|---|---|---|---|---|---|---|---|---|---|
| MambaIR | 0.7869 | 0.7956 | 0.7714 | 0.8445 | 0.8071 | 0.8240 | 3.57M | 19.71G | 29MB |
| CSMamba | 0.7140 | 0.7201 | 0.6934 | 0.8171 | 0.7503 | 0.7773 | 12.68M | 15.44G | 84MB |
| PlainMamba | 0.7787 | 0.7896 | 0.7617 | 0.8531 | 0.8064 | 0.8201 | 2.20M | **14.09G** | **18MB** |
| SCSegamba | 0.8204 | 0.8255 | **0.8241** | 0.8545 | 0.8390 | 0.8479 | 2.80M | 18.16G | 37MB |
| **SCRWKV (Ours)** | **0.8245** | **0.8313** | 0.8213 | **0.8655** | **0.8428** | **0.8512** | **1.22M** | 22.78G | 28MB |

*Table 9.* Comparison of different Mamba-based methods and our proposed method under the same number of core blocks ($L = 4$). Best results are highlighted in green and the second best are blue.

| Methods | ODS | OIS | P | R | F1 | mIoU | Params↓ | FLOPs↓ | Model Size↓ |
|---|---|---|---|---|---|---|---|---|---|
| UNet | 0.8142 | 0.8193 | **0.8301** | 0.8401 | 0.8351 | 0.8443 | 1.33M | 24.07G | 30MB |
| Ham | 0.8037 | 0.8113 | 0.8267 | 0.8247 | 0.8257 | 0.8376 | **1.22M** | 23.14G | **28MB** |
| SegFormer | 0.8164 | 0.8235 | 0.8238 | 0.8496 | 0.8365 | 0.8453 | **1.22M** | **22.67G** | **28MB** |
| MFS | 0.8087 | 0.8250 | 0.8158 | 0.8602 | 0.8374 | 0.8462 | **1.22M** | 23.00G | **28MB** |
| **CSHF (Ours)** | **0.8245** | **0.8313** | 0.8213 | **0.8655** | **0.8428** | **0.8512** | **1.22M** | 22.78G | **28MB** |

*Table 10.* Ablation study of different high-performance segmentation heads. Best results are in green, and second-best results are blue.

| AMCM | GBST | DSCD | ODS | OIS | P | R | F1 | mIoU | Params↓ | FLOPs↓ | Model Size↓ |
|---|---|---|---|---|---|---|---|---|---|---|---|
| ✓ | ✗ | ✗ | 0.8173 | 0.8236 | 0.8236 | 0.8486 | 0.8359 | 0.8461 | 1.21M | 22.51G | 27MB |
| ✗ | ✓ | ✗ | 0.8147 | 0.8240 | 0.8071 | 0.8639 | 0.8346 | 0.8450 | **0.76M** | **16.59G** | **23MB** |
| ✗ | ✗ | ✓ | 0.8109 | 0.8209 | 0.8075 | 0.8622 | 0.8340 | 0.8426 | 0.77M | 16.87G | 25MB |
| ✓ | ✓ | ✗ | 0.8170 | 0.8269 | 0.7900 | **0.8833** | 0.8340 | 0.8460 | 1.21M | 22.51G | 27MB |
| ✓ | ✗ | ✓ | 0.8183 | 0.8229 | **0.8327** | 0.8436 | 0.8381 | 0.8470 | 1.22M | 22.78G | 29MB |
| ✗ | ✓ | ✓ | 0.8061 | 0.8121 | 0.8294 | 0.8297 | 0.8295 | 0.8392 | 0.77M | 16.87G | 25MB |
| ✓ | ✓ | ✓ | **0.8245** | **0.8313** | 0.8213 | 0.8655 | **0.8428** | **0.8512** | 1.22M | 22.78G | 28MB |

*Table 11.* Ablation study of components within the SCRWKV. Best results are in green, and second-best results are blue.

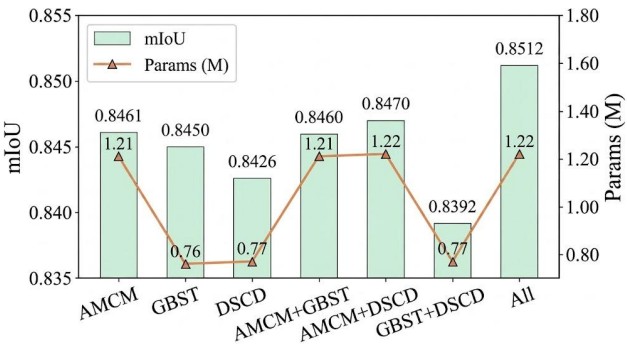

*Figure 9.* Comparison of SCRWKV internal components in terms of Params and mIoU.

possesses certain advantages in terms of parameters, computational cost, and model size under this configuration, its F1 score and mIoU were limited to 0.8064 and 0.8201, respectively, failing to achieve an ideal balance between performance and efficiency. In contrast, our method achieved superior comprehensive performance under the identical 4-layer VSS configuration. Specifically, with only 1.22M parameters and 22.78 GFLOPs, our model attained an F1

score of 0.8428 and an mIoU of 0.8512, significantly outperforming all compared Mamba-based methods. Compared to the second-best performing SCSegamba (Liu et al., 2025), our method further improved the F1 score and mIoU by 0.38% and 0.33%, respectively, while maintaining a more compact parameter scale and model size.

**Impact of Segmentation Heads.** To evaluate the effectiveness of the proposed lightweight decoder, we conduct a comparative analysis between the CSHF head and representative SOTA segmentation heads. As evidenced by the results in Table 10, our CSHF head achieves superior performance, yielding the highest mIoU of 0.8512 and an F1 score of 0.8428 among all compared methods. Crucially, CSHF achieves these gains while maintaining a remarkably compact footprint of 1.22M parameters, which matches the most lightweight architectures available. Although SegFormer exhibits marginally lower FLOPs, CSHF delivers significant accuracy improvements with negligible additional computational overhead. These findings demonstrate that CSHF achieves an optimal equilibrium between semantic aggregation capability and extreme parameter efficiency, making it suitable for resource-constrained devices.

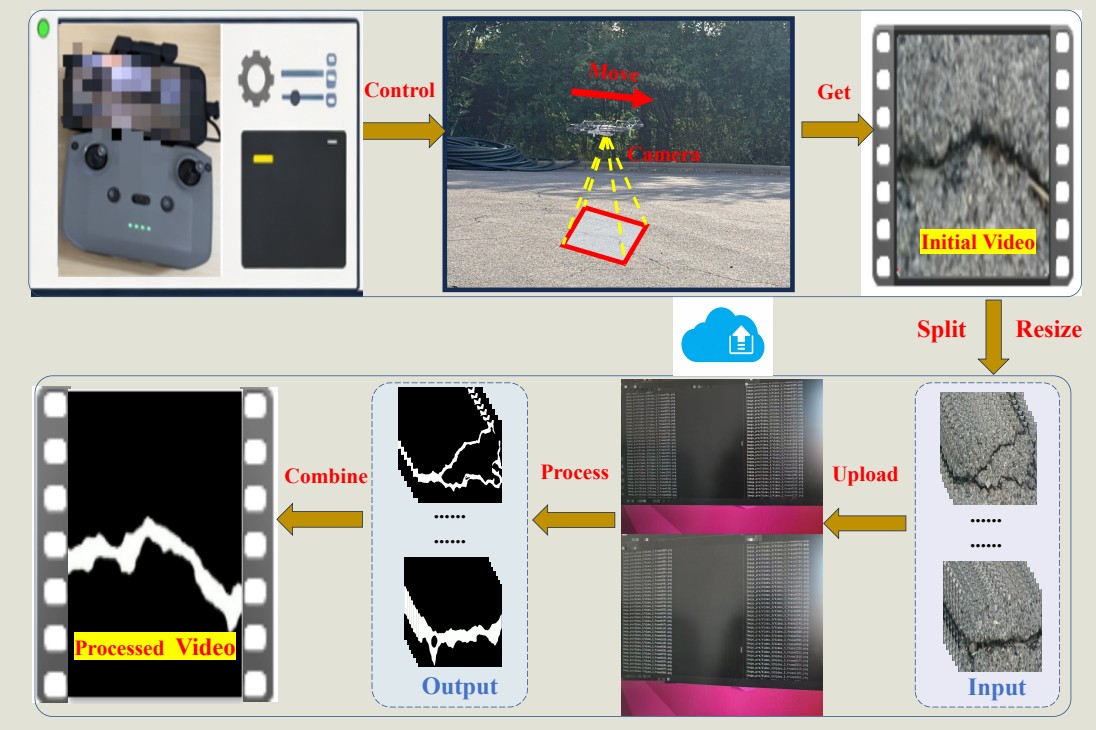

*Figure 10.* Practical deployment illustration. An intelligent UAV is deployed above outdoor road surfaces to perform low-altitude flight. The UAV is remotely controlled using a handheld controller in conjunction with a server terminal. During operation, the UAV continuously transmits real-time video data to the server, where the data are processed to generate the final outputs.

**Component Effectiveness Analysis.** To rigorously evaluate the individual contribution and necessity of each core component within the SFE backbone—namely the AMCM, GBST, and DSCD—we conducted a comprehensive "leave-one-out" ablation study. The empirical results, summarized in Table 11 and Figure 9, provide a clear quantitative justification for our architectural choices.Specifically, the removal of the GBST module precipitates a substantial decline in performance, underscoring its indispensable role in preserving and capturing global topological continuity. Likewise, the exclusion of either the AMCM or DSCD modules leads to a noticeable degradation in feature extraction efficiency, limiting the model's ability to resolve intricate patterns. The full SCRWKV configuration achieves superior performance across all metrics. This confirms that these components do not merely function in isolation but operate synergistically, forming a cohesive framework that significantly bolsters the precision of structural defect segmentation.

## 9 Real-world Deployment Applications

To validate the practical effectiveness of the proposed SCR-WKV, we executed a real-world deployment and compared its performance against other SOTA methods. Specifically, our experimental system comprises two primary components: an intelligent Unmanned Aerial Vehicle and a remote server. The UAV utilized is the DJI Mini SE, a device dis-

tinguished by its lightweight design and agility. It features a 3-axis mechanical stabilization gimbal and a 1/2.3-inch CMOS image sensor, enabling the capture of high-quality crack images. The remote server consists of a laptop workstation powered by an Intel Core i7-14650 CPU and running the Ubuntu 22.04 operating system. Communication between the UAV and the server is established via a wireless network. This setup emulates resource-constrained edge devices to evaluate the performance of SCRWKV in real-world aerial inspection scenarios.

During the actual deployment process, as illustrated in Figure 10, the UAV was operated in low-altitude flight above concrete pavement exhibiting widespread cracking. We remotely controlled the UAV to maintain a stable flight trajectory at a constant low cruising speed. The onboard camera captured a video stream at a frame rate of 30 frames per second. To optimize bandwidth for real-time inference, the recorded video data was resized to a resolution of $512 \times 512$ and transmitted to the server in real-time. Upon receiving the video data, the server first segmented it into individual frames and then fed each frame into the pre-trained SCRWKV model—which had been trained on all available datasets—for inference. Once segmentation was complete, the server reassembled the processed frames into a video to produce the final output. This setup simulates real-time crack segmentation in a production environment.

Furthermore, we deployed the pre-trained weights of other

| Methods | Year | FPS↑ | Inf Time ↓ |
|---|---|---|---|
| RIND | ICCV 2021 | 11 | 0.0909 |
| SFIAN | TITS 2023 | 35 | 0.0286 |
| CTCrackSeg | ICIP 2023 | 28 | 0.0357 |
| DTrCNet | AIC 2023 | **47** | **0.0213** |
| SimCrack | WACV 2024 | 29 | 0.0345 |
| Crackmer | AIC 2024 | 31 | 0.0323 |
| MambaIR | ECCV 2024 | 25 | 0.0400 |
| CSMamba | arXiv 2024 | 16 | 0.0625 |
| PlainMamba | BMVC 2024 | 6 | 0.1667 |
| SCSegamba | CVPR 2025 | 32 | 0.0313 |
| **SCRWKV (Ours)** | ICML 2026 | 46 | 0.0216 |

*Table 12.* Comparison of inference speed (FPS) and inference time with 10 SOTA methods on resource-constrained server. Best results are in **green**, and second-best results are blue.

SOTA methods on the server for comparative analysis. As presented in Table 12, our SCRWKV achieved an inference speed of 0.0216 seconds per frame on the resource-constrained server, a performance virtually on par with the fastest competitor, DTrCNet (Xiang et al., 2023). This demonstrates that our method possesses exceptional real-time capabilities, making it highly suitable for real-time crack segmentation in video data.

As illustrated in Figure 11, compared to current SOTA methods, SCRWKV demonstrates superior stability and noise immunity in dynamic video scenarios, generating highly coherent crack segmentation maps. While advanced Mamba-based models such as SCSegamba (Liu et al., 2025) and MambaIR (Guo et al., 2024) achieve impressive metrics on static benchmarks, they occasionally exhibit temporal instability when processing video streams, making them susceptible to misclassifying dynamic environmental artifacts—such as moving shadows or water stains—as structural defects. Similarly, CNN-based methods such as RIND (Pu et al., 2021) and DTrCNet (Xiang et al., 2023), despite possessing rapid inference speeds, often falter in maintaining topological continuity, resulting in cracks appearing as disconnected fragments within the generated maps. Consequently, as evidenced by the robust performance during practical UAV deployment, our SCRWKV framework sustains exceptional segmentation fidelity while operating under an ultra-lightweight constraint of only 1.22M parameters. Even when subjected to rapid motion blur or complex lighting fluctuations, the model effectively preserves topological continuity without incurring significant computational overhead. These results conclusively validate that the proposed architecture successfully bridges the gap between high-precision crack identification and real-time edge deployment efficiency, establishing a new benchmark for

**Algorithm 1** GBST execution process

1: **Input:** Feature matrix $\mathbf{X} \in \mathbb{R}^{B \times N \times C}$, shift pixel $s$, patch resolution $(H, W)$
2: **Output:** Transformed feature matrix $\mathbf{Y} \in \mathbb{R}^{B \times N \times C}$
3: **Initialize:** $\mathbf{X}_{res} \leftarrow \text{Reshape}(\mathbf{X}, [B, C, H, W])$, $\mathbf{Y}_{res} \leftarrow \mathbf{0} \in \mathbb{R}^{B \times C \times H \times W}$
4: $C_{half} \leftarrow \lfloor C/2 \rfloor, C_q \leftarrow \lfloor C_{half}/4 \rfloor$
5: {— Part I: Outward Diffusion (Expansion Stream) —}
6: **for** $k \leftarrow 1$ **to** 4 **do**
7:    $start, end \leftarrow (k-1) \cdot C_q, k \cdot C_q$
8:    **if** $k = 1$ **then**
9:      $\mathbf{Y}_{res}[:, start : end, :, s : W] \leftarrow \mathbf{X}_{res}[:, start : end, :, 0 : W - s]$ {Right shift}
10:    **else if** $k = 2$ **then**
11:      $\mathbf{Y}_{res}[:, start : end, :, 0 : W - s] \leftarrow \mathbf{X}_{res}[:, start : end, :, s : W]$ {Left shift}
12:    **else if** $k = 3$ **then**
13:      $\mathbf{Y}_{res}[:, start : end, s : H, :] \leftarrow \mathbf{X}_{res}[:, start : end, 0 : H - s, :]$ {Down shift}
14:    **else**
15:      $\mathbf{Y}_{res}[:, start : C_{half}, 0 : H - s, :] \leftarrow \mathbf{X}_{res}[:, start : C_{half}, s : H, :]$ {Up shift}
16:    **end if**
17: **end for**
18: {— Part II: Inward Convergence (Shrink Stream) —}
19: $S_{idx} \leftarrow C_{half}, C_{sq} \leftarrow \lfloor (C - C_{half})/4 \rfloor$
20: **for** $k \leftarrow 1$ **to** 4 **do**
21:    $start, end \leftarrow S_{idx} + (k-1) \cdot C_{sq}, S_{idx} + k \cdot C_{sq}$
22:    **if** $k = 1$ **then**
23:      $\mathbf{Y}_{res}[:, start : end, :, 0 : W - s] \leftarrow \mathbf{X}_{res}[:, start : end, :, s : W]$ {Opposite Left}
24:    **else if** $k = 2$ **then**
25:      $\mathbf{Y}_{res}[:, start : end, :, s : W] \leftarrow \mathbf{X}_{res}[:, start : end, :, 0 : W - s]$ {Opposite Right}
26:    **else if** $k = 3$ **then**
27:      $\mathbf{Y}_{res}[:, start : end, 0 : H - s, :] \leftarrow \mathbf{X}_{res}[:, start : end, s : H, :]$ {Opposite Up}
28:    **else**
29:      $\mathbf{Y}_{res}[:, start : C, s : H, :] \leftarrow \mathbf{X}_{res}[:, start : C, 0 : H - s, :]$ {Opposite Down}
30:    **end if**
31: **end for**
32: **Return** $\mathbf{Y} \leftarrow \text{Flatten}(\mathbf{Y}_{res}, [B, N, C])$

automated structural health monitoring in dynamic environments.

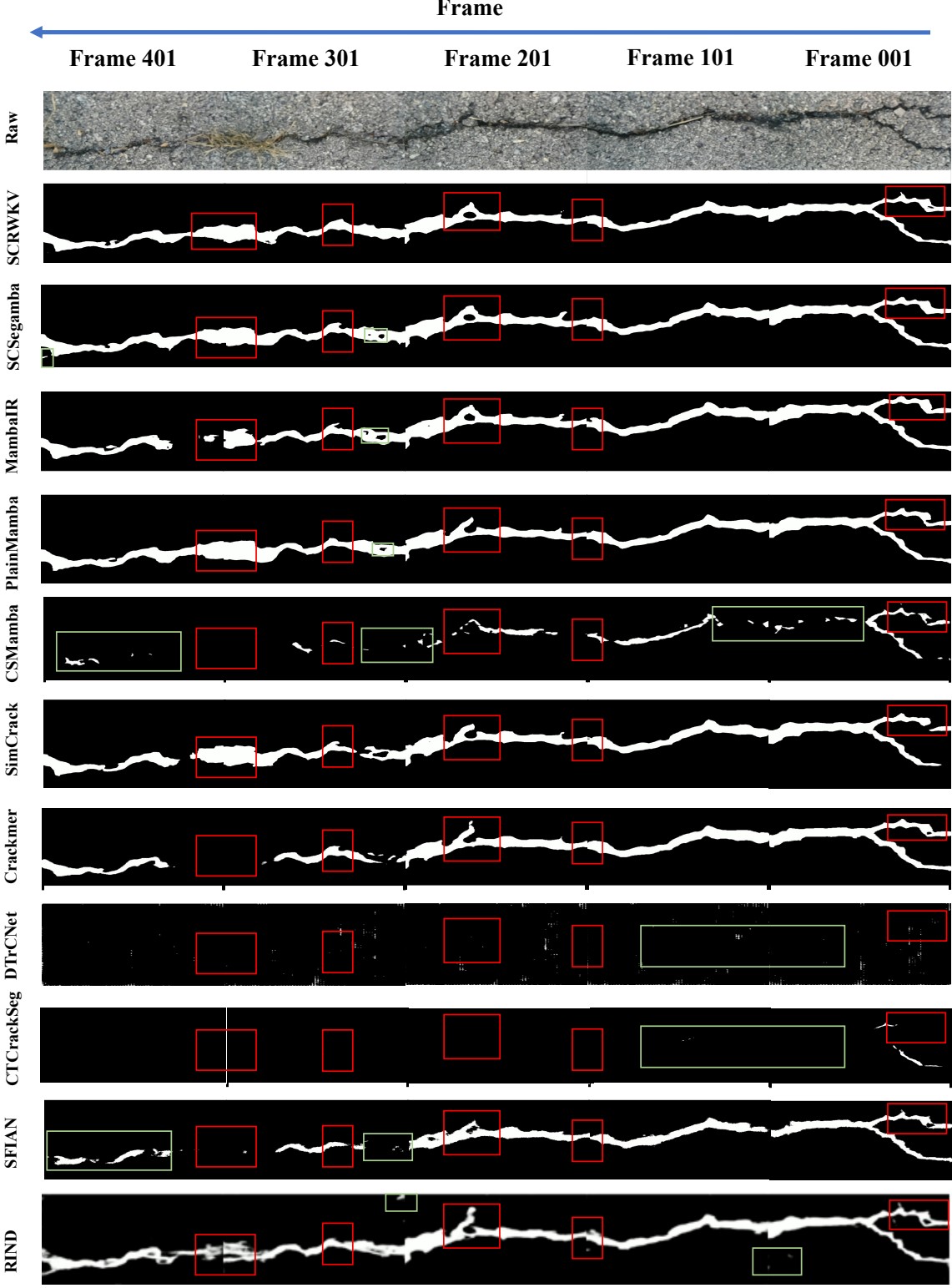

*Figure 11.* Visualisation comparison on video data keyframes. The interval between keyframes is 100 frames in order to ensure continuity of observation. Red boxes highlight critical details, and green boxes mark misidentified regions.

