# OpenReview forum: "SCRWKV: Ultra-Compact Structure-Calibrated Vision-RWKV for Topological Crack Segmentation"
_ICML.cc/2026/Conference — ICML 2026 regular_

### Official Review · Reviewer_Zzdb · 2026-03-02

**Soundness:** 3
**Presentation:** 3
**Significance:** 2
**Originality:** 3
**Overall Recommendation:** 4
**Confidence:** 2

**Summary:**

This work proposes SCRWKV, which is a light neural network trained for crack segmentation. SCRWKV only contains 1.22M params and is the second fastest compared with various baselines. SCRWKV includes a structure-field encoder and a cross-scale harmonic fusion decoder. Experiments on four commonly used datasets demonstarte the SOTA performance of SCRWKV, beating many representative baselines like SCSegamba.

**Compliance With Llm Reviewing Policy:**

Affirmed.

**Final Justification:**

Thanks the authors for their time and effort. This rebuttal fully addressed most of my concerns, except the second one about EfficientCrackNet. I will remain my positive recommendation.

**Key Questions For Authors:**

- Why the FLOPs ranks third but the latency ranks the second and is very close to the first? Is the latencies of all methods tested under the same condition?
- Why the baselines methods are different in this work and EfficientCrackNet [1]?
- Do you consider adding more experiments on CrackSeg9k[2] and NHA12D[3], to better demonstrate your better performance?

References
- [1] EfficientCrackNet: A Lightweight Model for Crack Segmentation. Abid Hasan Zim, Aquib Iqbal, Zaid Al-Huda, Asad Malik, Minoru Kuribayash. WACV 2025.
- [2] CrackSeg9k: A Collection and Benchmark for Crack Segmentation Datasets and Frameworks. Shreyas Kulkarni, Shreyas Singh, Dhananjay Balakrishnan, Siddharth Sharma, Saipraneeth Devunuri, Sai Chowdeswara Rao Korlapati. ECCV 2022 Workshops.
- [3]NHA12D: A New Pavement Crack Dataset and a Comparison Study Of Crack Detection Algorithms. Zhening Huang, Weiwei Chen, Abir Al-Tabbaa, Ioannis Brilakis. EC3 2022.

**Limitations:**

There is an impact statement but there is no limitation discussion. For potential suggestions, the zero-shot generalization ability on in-the-wild scenarios may worth consideration, because SCRWKV is still trained and tested under in-domain settings.

**Strengths And Weaknesses:**

Strengths
- This work studies an important problem: balancing the performance and computational efficiency in crack segmentation.
- The RWKV-based design is interesting and emperically works well.
- The experiments is comperhensive, showing a clear performance gain over the previous best method.

Weaknesses
- The authors claimed that their model exhibits *linear complexity*, but lacks a theoretical complexity analysis.
- Other concerns are detailed in "Key Questions For Authors".

---

> ### Author Rebuttal · Authors · 2026-03-30
>
> We sincerely appreciate the reviewer for the constructive feedback!
> W stands for Weakness, Q stands for Question. We address each point below.
>
> # About W1:
> Theoretical complexity analysis for each component. Let N = H x W denote spatial tokens, C the channel dimension.
>
> GBST: O(N*C). Parameter-free channel-wise shifts (Eq. 9-10). Bidirectional design is a constant factor.
>
> AMCM: O(N*C). Depthwise convolutions with bounded kernels K in {5,7,9,11} (Eq. 5). Spatial attention (Eq. 7) on fixed G x G grid is negligible.
>
> Dy-WKV: O(N*C). Recurrent formulation (Eq. 14) reduces per-token cost to O(C), unlike Transformer's O(N^2*C). DSCD (Eq. 12) adds only element-wise operations.
>
> CSHF: O(N*C). Attention (Eq. 21) over 4 fixed scale levels, not N tokens.
>
> Overall: O(N*C), strictly linear. Empirically, a 16× resolution increase yields only 1.73× AMCM latency and 1.40× CSHF latency, as detailed in Q1.
>
> # About Q1:
>
> Yes, all measurements used identical conditions: same server, 512x512 input, batch size 1. Details in Section 9.
>
> The FLOPs-latency discrepancy reflects practical hardware behavior. FLOPs measure theoretical arithmetic under idealized conditions, whereas actual speed depends on memory access patterns, operator parallelism, and cache utilization. Crackmer achieves only 31 FPS despite its lower 14.94G FLOPs, as its dual-path CNN-Transformer hybrid introduces fragmented memory access and heterogeneous module routing. SCSegamba reaches just 32 FPS at 18.16G FLOPs because irregular sequential scanning disrupts memory locality and underutilizes hardware parallelism. In contrast, SCRWKV's operations are hardware-friendly primitives with sequential memory access and high data reuse, achieving 46 FPS, merely 1.4% slower than the fastest DTrCNet despite ranking third in theoretical FLOPs.
>
> # About Q2:
>
> We thank the reviewer for pointing out EfficientCrackNet. While it is a highly relevant concurrent work, our baseline selection diverges from EfficientCrackNet, driven by three key evaluation criteria. Temporal relevance: we prioritize methods from 2023 to 2025 such as SCSegamba at CVPR 2025 and MambaIR at ECCV 2024. Architectural coverage: 10 baselines spanning CNN, Transformer, and Mamba families. Fairness: all baselines retrained from scratch under identical splits, hyperparameters, and hardware.
> # About Q3:
>
> We conducted additional experiments on both CrackSeg9k and NHA12D to further demonstrate the superiority of SCRWKV. Visualizations at:[Link1](https://anonymous.4open.science/r/anonym-3EB5/data-com.pdf).
>
> CrackSeg9k :
>
> | Method     | mIoU   | ODS    | OIS    | F1     |
> |-|-|-|-|-|
> | Crackmer   | 0.6129 | 0.4721 | 0.4981 | 0.2424 |
> | SCSegamba  | 0.6297 | 0.4557 | 0.4748 | 0.3099 |
> | MambaIR    | 0.6432 | 0.4903 | 0.5051 | 0.3145 |
> | PlainMamba | 0.6434 | 0.4548 | 0.4700 | 0.4170 |
> | RIND       | 0.6787 | 0.5453 | 0.2858 | 0.5453 |
> | SCRWKV     | 0.6889 | 0.5240 | 0.5474 | 0.6125 |
>
> NHA12D :
>
> | Method     | mIoU   | ODS    | OIS    | F1     |
> |-|-|-|-|-|
> | Crackmer   | 0.4833 | 0.0531 | 0.0544 | 0.0093 |
> | RIND       | 0.5503 | 0.2180 | 0.2515 | 0.2180 |
> | PlainMamba | 0.5988 | 0.3332 | 0.3376 | 0.3429 |
> | MambaIR    | 0.6189 | 0.3849 | 0.3912 | 0.3932 |
> | SCSegamba  | 0.6517 | 0.4602 | 0.4780 | 0.4543 |
> | SCRWKV     | 0.7117 | 0.5902 | 0.6016 | 0.5837 |
>
> SCRWKV ranks first on both datasets. On CrackSeg9k, mIoU improves by +1.02% over runner-up RIND, and F1 improves by +6.72%, demonstrating strong robustness under heterogeneous data distributions from 12 different sources. On NHA12D, the advantage is more pronounced: +6.00% mIoU and +12.94% F1 over SCSegamba. Notably, many competing methods degrade severely on this challenging highway dataset, while SCRWKV maintains stable performance, validating DSCD's effectiveness in suppressing complex background noise.
>
> # About Limitations:
>
> We deployed pre-trained SCRWKV, trained only on benchmark datasets, on a DJI Mini SE UAV for real-time crack segmentation of previously unseen outdoor pavement, without any fine-tuning or domain adaptation. The UAV captured video at 30 FPS and transmitted frames to a resource-constrained server. Video demonstrations are available at [Link2](https://anonymous.4open.science/r/anonym-3EB5/real-word.gif) and [Link3](https://anonymous.4open.science/r/anonym-3EB5/compared_model_vedio.mp4).
>
> SCRWKV generates coherent and stable segmentation maps under varying illumination, motion blur, and unseen textures. In contrast, competing methods exhibit notable failure modes: Crackmer produces artifacts with flickering, CSMamba misclassifies shadows as cracks, and CNN-based methods produce topologically fragmented results. This zero-shot robustness is attributable to the synergistic design of GBST, DSCD, and AMCM. We acknowledge that systematic zero-shot cross-domain evaluation remains a valuable future direction. The full deployment analysis is in Section 9. We hope these additions effectively address your concerns.

---

> > ### Author Rebuttal · Reviewer_Zzdb · 2026-04-03
> >
> > Thanks the authors for their time and effort. This rebuttal fully addressed most of my concerns, except the second one about EfficientCrackNet. I will remain my positive recommendation.

---

> > > ### Author Response · Authors · 2026-04-03
> > >
> > > Thank you again for your engagement and for maintaining your positive recommendation despite this remaining concern.
> > >
> > > Regarding EfficientCrackNet, we understand this point was not fully resolved. To briefly reiterate: while it is a highly relevant concurrent work, our baseline selection was guided by temporal relevance, architectural coverage, and fairness of retraining conditions, criteria under which our chosen baselines were deemed most appropriate. We hope our explanation provides sufficient context for this design choice.
> > >
> > > We would like to express our heartfelt gratitude for your time, patience, and thorough engagement throughout the entire review process. Your insightful comments and constructive feedback have been truly invaluable to us, and we deeply appreciate the care and effort you have dedicated to reviewing our work.

---

### Official Review · Reviewer_pmTD · 2026-03-10

**Soundness:** 3
**Presentation:** 4
**Significance:** 3
**Originality:** 3
**Overall Recommendation:** 4
**Confidence:** 4

**Summary:**

This paper proposes SCRWKV, an ultra-compact vision model for topological crack segmentation. The method builds on the Vision-RWKV architecture and introduces structural modules to better capture irregular crack geometries and long-range dependencies. Specifically, the model employs a Structure-Field Encoder (SFE) backbone with an Adaptive Multi-scale Cascaded Modulator (AMCM) to initialize geometric correlations, along with a Structure-Calibrated Insight Unit (SCIU) that integrates spatial mixing, channel mixing, and topology-aware mechanisms. These components improve robustness to noise and help preserve the continuity of thin crack structures in complex backgrounds.
Experiments on several crack segmentation datasets show that SCRWKV achieves strong segmentation performance while remaining highly lightweight and suitable for real-time inference.

**Compliance With Llm Reviewing Policy:**

Affirmed.

**Key Questions For Authors:**

1. Generalization to other segmentation tasks. Have the authors evaluated SCRWKV on other topology-sensitive tasks such as vessel segmentation, road extraction, or skeleton prediction to demonstrate the broader applicability of the architecture?

2. Robustness to extreme noise or domain shifts. How does the model perform when evaluated on datasets with significantly different imaging conditions or crack types, such as industrial defects or concrete cracks under varying lighting conditions?

**Limitations:**

Yes

**Strengths And Weaknesses:**

Strengths

-	Lightweight yet effective architecture. The proposed SCRWKV achieves competitive segmentation accuracy with only about 1.22M parameters, making it suitable for resource-constrained or real-time applications such as UAV-based crack inspection.

-	Architecture tailored to crack topology. The model introduces specialized modules (AMCM, SCIU) designed to capture long, thin, and irregular crack structures, addressing the limitations of existing methods that flatten spatial structure or treat all tokens equally.

-	Strong empirical validation. Experiments on multiple public crack datasets show improvements in segmentation quality, including better preservation of topological continuity and robustness to noise compared with CNN and state-space baselines.

Weaknesses

-	Domain-specific evaluation. Experiments are restricted to crack segmentation datasets. It remains unclear whether the proposed architecture generalizes to other structured segmentation tasks (e.g., vessels, roads, or filaments).

-	Limited analysis of RWKV advantages. Although the work builds on Vision-RWKV, the paper provides limited empirical analysis comparing RWKV-based designs against stronger transformer or state-space baselines with similar parameter counts.

-	Insufficient discussion of failure cases. The paper focuses primarily on qualitative improvements but offers limited analysis of scenarios where the model fails, such as extremely noisy scenes or severe illumination artifacts.

---

> ### Author Rebuttal · Authors · 2026-03-30
>
> We sincerely appreciate the reviewer for the constructive feedback!
> W stands for Weakness, Q stands for Question. We address each point below.
>
> # About W1 & Q1:
> We evaluated SCRWKV on two additional topology-sensitive tasks beyond crack segmentation. Visualizations of all cross-domain results are available at:[Link1](https://anonymous.4open.science/r/anonym1-B2FC/comparison/two.pdf)
>
> MEMO (Vessel Segmentation):
> | Method        | mIoU   | ODS    | OIS    | F1     |
> |-|-|-|-|-|
> | Crackmer      | 0.4690 | 0.1173 | 0.1174 | 0.1173 |
> | PlainMamba    | 0.6226 | 0.4742 | 0.4742 | 0.4763 |
> | RIND          | 0.6221 | 0.4832 | 0.4922 | 0.4832 |
> | MambaIR       | 0.6362 | 0.5106 | 0.5107 | 0.5124 |
> | SCSegamba     | 0.6412 | 0.5183 | 0.5183 | 0.5192 |
> | SCRWKV (Ours) | 0.6610 | 0.5546 | 0.5558 | 0.5541 |
>
> SCRWKV surpasses SCSegamba by +3.09% in mIoU and +6.72% in F1. It achieves first place across all metrics, suggesting GBST captures general topological patterns shared across tubular structures.
>
> KUST4K (Road Extraction):
> | Method        | mIoU   | ODS    | OIS    | F1     |
> |-|-|-|-|-|
> | Crackmer      | 0.6953 | 0.8292 | 0.8573 | 0.8424 |
> | PlainMamba    | 0.7658 | 0.8808 | 0.9017 | 0.8889 |
> | SCSegamba     | 0.7695 | 0.8839 | 0.9087 | 0.8704 |
> | RIND          | 0.7725 | 0.8856 | 0.8877 | 0.8856 |
> | MambaIR       | 0.7899 | 0.8973 | 0.9137 | 0.9038 |
> | SCRWKV (Ours) | 0.8013 | 0.9062 | 0.9164 | 0.9116 |
>
> SCRWKV surpasses MambaIR by +1.44% in mIoU and +0.86% in F1. These results confirm strong generalization to diverse topology-sensitive tasks including vessel and road structures, validating that SCRWKV's architectural contributions extend beyond crack-specific scenarios.
>
> # About W2:
> To further validate our RWKV-based design, we compare with Vision-RWKV and MobileViT under identical settings to isolate the contribution of our proposed modules.
>
> TUT Dataset:
>
> | Model         | ODS    | OIS    | F1     | mIoU   |
> |-|-|-|-|-|
> | MobileViT     | 0.6970 | 0.7149 | 0.6147 | 0.7728 |
> | Vision-RWKV   | 0.7625 | 0.7688 | 0.7849 | 0.8091 |
> | SCRWKV (Ours) | 0.8245 | 0.8313 | 0.8428 | 0.8512 |
>
> DeepCrack Dataset:
>
> | Model         | mIoU    | F1    | Params (M) | FLOPs (G) |
> |-|-|-|-|-|
> | MobileViT     | 0.8890 | 0.8970 | 4.86        |  7.96    |
> | Vision-RWKV   | 0.8806 | 0.8858 | 5.96       |  6.35     |
> | SCRWKV (Ours) | 0.9289 | 0.9363 | 1.22       | 22.78     |
>
> On TUT, SCRWKV significantly boosts Vision-RWKV by +5.79% F1 and +4.21% mIoU, demonstrating the effectiveness of our proposed modules GBST, DSCD, and AMCM on top of the RWKV backbone. This progressive improvement from MobileViT to Vision-RWKV to SCRWKV illustrates that the gains arise from both the RWKV foundation and our structure-calibrated enhancements. On DeepCrack, SCRWKV achieves the highest mIoU and F1 across all models with only 1.22M parameters, 3.64M fewer than MobileViT and 4.74M fewer than Vision-RWKV. This confirms that our structural calibration achieves superior performance with substantially better parameter efficiency compared to Transformer-based lightweight baselines.
>
> # About W3 & Q2:
> TUT already encompasses diverse imaging conditions. To further validate under distinct industrial scenarios, we tested on SteelCrack. Visualizations of results are available at:[Link2](https://anonymous.4open.science/r/anonym1-B2FC/comparison/Steel.pdf)
>
> SteelCrack (Industrial Defect):
>
> | Method     | ODS    | OIS    | F1     | mIoU   |
> |-|-|-|-|-|
> | CSMamba    | 0.6839 | 0.6874 | 0.7517 | 0.7666 |
> | CTCrackSeg | 0.7301 | 0.7393 | 0.7883 | 0.8009 |
> | Crackmer   | 0.7332 | 0.7448 | 0.7691 | 0.7990 |
> | RIND       | 0.7487 | 0.8092 | 0.7752 | 0.8145 |
> | MambaIR    | 0.8013 | 0.8048 | 0.8465 | 0.8412 |
> | SCSegamba  | 0.8056 | 0.8103 | 0.8363 | 0.8468 |
> | PlainMamba | 0.8104 | 0.8137 | 0.8547 | 0.8467 |
> | DTrCNet    | 0.8089 | 0.8276 | 0.8436 | 0.8484 |
> | SimCrack   | 0.8200 | 0.8337 | 0.8570 | 0.8552 |
> | SCRWKV (Ours)    | 0.8578 | 0.8641 | 0.8899 | 0.8820 |
>
> SCRWKV surpasses the runner-up SimCrack by +3.29% in F1 and +2.68% in mIoU, confirming robust generalization to industrial defect scenarios with fundamentally different surface properties. The performance gap on SteelCrack is notably larger than on original benchmarks, suggesting topology-aware modules provide even greater advantages under novel imaging conditions. SCRWKV's design is inherently material-agnostic: GBST models geometric structure rather than material-specific textures, and DSCD filters noise independent of source, enabling transferability across diverse inspection scenarios.
>
> Regarding failure cases, SCRWKV occasionally produces false positives in regions with strong texture patterns resembling crack morphology, such as concrete joints and painted road markings. Under extreme low-contrast conditions, the model may produce slightly incomplete segmentation. These failure modes are shared across all compared methods but represent valuable future directions. We hope these additions address your concerns.

---

> > ### Author Rebuttal · Reviewer_pmTD · 2026-04-03
> >
> > The rebuttal has addressed my concerns, so I maintain my initial rating (weak accept).

---

> > > ### Author Response · Authors · 2026-04-04
> > >
> > > Thank you for your acknowledgement and for maintaining your positive initial rating. Your insightful questions truly helped us clarify our contributions, and we are very glad that our responses provided the necessary explanations.
> > >
> > > Thank you again for your time!

---

### Official Review · Reviewer_bdqW · 2026-03-12

**Soundness:** 3
**Presentation:** 4
**Significance:** 3
**Originality:** 3
**Overall Recommendation:** 5
**Confidence:** 5

**Summary:**

This paper proposes a very lightweight visual rwkv architecture (scrwkv), which improves the spatial modeling and segmentation performance of fine-grained topological cracks by introducing geometry guided bidirectional structure transformation (gbst) and dynamic self calibration attenuation (DSCD).

**Compliance With Llm Reviewing Policy:**

Affirmed.

**Final Justification:**

This paper proposed a novel topological crack segmentation method based on RWKV architecture. At the rebuttal stage, the author answered my questions one by one and provide several solid extra experiments and visualization maps to prove his opinion. Thus, I raise my score to 5.

**Key Questions For Authors:**

1. At the beginning of the paper, it is emphasized to inherit the linear complexity advantage of rwkv, but in the actual feature extraction and fusion stage, a large number of complex local operations are introduced. For example, the AMCM uses large deep separable convolution up to $11\times11$ and the CSHF decoder performs intensive cross-scale channel fusion. When processing high-resolution input, these high memory access costs (MAC) are likely to offset the efficiency dividends brought by sequence modeling.

2. When comparing different spatial interaction mechanisms, such as Q-shift and GBST, the effective receptive field of each branch was not strictly aligned. In essence, GBST implicitly expands the local receptive field by four-way characteristic flow. The improvement of model performance is likely to come from the undifferentiated expansion of the receptive field, rather than the "accurate capture of topological manifold" claimed by the author.

**Limitations:**

Yes

**Strengths And Weaknesses:**

# Strengths
This paper shows a very high engineering level in the balance between parameter compression and model efficiency. It can surpass most existing Mamba and transformer architectures with a large number of parameters on multiple fracture segmentation benchmarks with only 1.22m parameters. In addition, it is an enlightening improvement of spatial interaction to abandon the rigid Q-shift mechanism of traditional visual rwkv and adopt bidirectional structure transformation (gbst) designed for topological continuity.

# Weaknesses
1. There is a serious field mismatch between this paper and the core positioning of ICML. ICML favors breakthroughs in basic machine learning theory or architecture innovation with broad paradigm significance.

2. The validation of the generalization ability of the model is extremely limited, which weakens its theoretical persuasion as a "new architecture". This paper only tests the macro crack data sets of four specific materials such as crack500. If scrwkv claims that it has the advantages of pervasive topology awareness and long-range modeling, it must be verified in more challenging general intensive prediction tasks, or in complex scenes with high precision, high generalization requirements and sparse data distribution.

3. The author wants to prove the dominance under very low parameters, not only compared with the Transformer or the Mamba model. You must directly compete with the current top lightweight dense prediction network, such as the latest variant of edgenext or mobilevit, and other rwkv models optimized for vision.

---

> ### Author Rebuttal · Authors · 2026-03-30
>
> We sincerely appreciate the reviewer for the constructive feedback!
> W stands for Weakness, Q stands for Question. We address each point below.
>
> # About W1:
> We respectfully disagree with the field mismatch concern. While crack segmentation is the primary evaluation domain, our core contributions address fundamental limitations in Vision-RWKV architectures at the backbone design level: GBST replaces rigid Q-Shift with adaptive bidirectional spatial interaction, DSCD introduces content-aware dynamic decay into linear recurrence, and AMCM compensates linear attention's local granularity limitation. These are general vision modeling improvements, not domain-specific engineering.
>
> Cross-domain experiments confirm this: SCRWKV achieves first place on MEMO for retinal vessel segmentation, KUST4K for road extraction, and SteelCrack for industrial defect detection, all surpassing existing methods by significant margins. Crack segmentation serves as a rigorous case study validating these architectural principles, which we believe fall within ICML's scope of efficient linear-complexity vision backbone design.
>
> # About W2:
> Three groups of cross-domain experiments. Visualizations at:[Link1](https://anonymous.4open.science/r/anonym1-B2FC/comparison/data.pdf)
>
> MEMO (Vessel Segmentation):
>
> | Method     | mIoU   | ODS    | OIS    | F1     |
> |-|-|-|-|-|
> | Crackmer   | 0.4690 | 0.1173 | 0.1174 | 0.1173 |
> | PlainMamba | 0.6226 | 0.4742 | 0.4742 | 0.4763 |
> | RIND       | 0.6221 | 0.4832 | 0.4922 | 0.4832 |
> | MambaIR    | 0.6362 | 0.5106 | 0.5107 | 0.5124 |
> | SCSegamba  | 0.6412 | 0.5183 | 0.5183 | 0.5192 |
> | Ours       | 0.6610 | 0.5546 | 0.5558 | 0.5541 |
>
> SCRWKV surpasses SCSegamba by 3.09% in mIoU and 6.72% in F1.
>
> KUST4K (Road Extraction):
>
> | Method     | mIoU   | ODS    | OIS    | F1     |
> |-|-|-|-|-|
> | Crackmer   | 0.6953 | 0.8292 | 0.8573 | 0.8424 |
> | PlainMamba | 0.7658 | 0.8808 | 0.9017 | 0.8889 |
> | SCSegamba  | 0.7695 | 0.8839 | 0.9087 | 0.8704 |
> | RIND       | 0.7725 | 0.8856 | 0.8877 | 0.8856 |
> | MambaIR    | 0.7899 | 0.8973 | 0.9137 | 0.9038 |
> | Ours       | 0.8013 | 0.9062 | 0.9164 | 0.9116 |
>
> SCRWKV surpasses MambaIR by 1.44% in mIoU and 0.86% in F1.
>
> SteelCrack (Industrial Defect):
>
> | Method     | mIoU   | ODS    | OIS    | F1     |
> |-|-|-|-|-|
> | Crackmer   | 0.7990 | 0.7332 | 0.7448 | 0.7691 |
> | PlainMamba | 0.8467 | 0.8104 | 0.8137 | 0.8547 |
> | SCSegamba  | 0.8468 | 0.8056 | 0.8103 | 0.8363 |
> | MambaIR    | 0.8412 | 0.8013 | 0.8048 | 0.8465 |
> | Ours       | 0.8820 | 0.8578 | 0.8641 | 0.8899 |
>
> SCRWKV surpasses SCSegamba by +3.52% mIoU and +5.36% F1 on steel surfaces with specular reflections.
>
> # About W3:
> Direct comparisons with top lightweight networks on DeepCrack:
>
> | Model      | mIoU   | F1     | FLOPs (G) | Params (M) |
> |-|-|-|-|-|
> | MobileViT  | 0.8890 | 0.8970 |     7.96  | 4.86  |
> | EdgeNeXt   | 0.9177 | 0.9198 |    5.37     |  5.30   |
> | Vision-RWKV   | 0.8806 | 0.8858 |   6.35  |  5.96      |
> | Ours       | 0.9289 | 0.9363 |   22.78|  1.22|
>
> SCRWKV outperforms EdgeNeXt by 1.12% mIoU with only 23.02% of its parameters, and outperforms MobileViT by 3.99% mIoU.
>
> We acknowledge higher FLOPs due to AMCM's multi-scale large-kernel operations. However, SCRWKV achieves 46 FPS. For edge devices, memory and storage are the primary bottlenecks. SCRWKV's 1.22M parameters and 28MB model size are the smallest among all methods, offering more practical deployment advantage than lower FLOPs with larger memory footprint.
> # About Q1:
> Wall-clock latency of AMCM and CSHF across resolutions:
>
> | Module | 512x512 | 1024x1024 | 2048x2048 |
> |-|-|-|-|
> | AMCM   | 0.006s  | 0.007s    | 0.010s    |
> | CSHF   | 0.048s  | 0.059s    | 0.067s    |
>
> AMCM scales sub-linearly: only 1.73x latency increase for 16x resolution growth. CSHF similarly scales with 1.40x increase. This confirms the theoretical MAC overhead does not translate into practical efficiency degradation.
> # About Q2:
> ERF analysis to disentangle receptive field expansion from topological modeling, under identical 4-layer encoder at 128x128. Visualizations at:[Link2](https://anonymous.4open.science/r/anonym1-B2FC/comparison/heatmap.pdf).
>
> | Method    | Area | Concentration | Entropy |
> |--|-|-|-|
> | Q-Shift   | 22        | 1.000         | 1.743        |
> | OmniShift | 36        | 0.866         | 2.006        |
> | Ours      | 24        | 1.000         | 1.850        |
>
> OmniShift has the largest ERF yet underperforms GBST in Table 5, refuting the hypothesis that gains stem from receptive field size. GBST achieves the highest concentration score among all three methods, confirming gradient energy focuses on structurally relevant regions rather than diffusing uniformly. GBST shows distinctive bidirectional peaks, fundamentally different from OmniShift's uniform spread.
>
> Feature activation heatmaps corroborate this: GBST activations concentrate along crack trajectories with continuity at bifurcations; Q-Shift produces fragmented activations; OmniShift introduces background noise.

---

> > ### Author Rebuttal · Reviewer_bdqW · 2026-04-04
> >
> > Thanks a lot for your concrete response! I decide improve my score to 5.

---

> > > ### Author Response · Authors · 2026-04-04
> > >
> > > Thank you so much for your kind words and for raising your score!!! We are thrilled to hear that our rebuttal has fully addressed your concerns.
> > >
> > > Your constructive comments and rigorous feedback throughout this process have been incredibly valuable, and they have genuinely helped us improve the quality and clarity of our work. We deeply appreciate the time, effort, and expertise you dedicated to reviewing our paper.
> > >
> > > Thanks again for your time and support!!!

---

### Official Review · Reviewer_wrEJ · 2026-03-17

**Soundness:** 3
**Presentation:** 3
**Significance:** 3
**Originality:** 2
**Overall Recommendation:** 4
**Confidence:** 3

**Summary:**

The paper proposes SCRWKV, a compact segmentation network designed for accurate crack detection under complex backgrounds and resource constraints. The work builds upon Vision-RWKV architectures and aims to address their limitations in modeling irregular crack topologies and handling noise-dominated scenes.

To this end, the authors introduce a Structure-Field Encoder (SFE) backbone composed of stacked Structure-Calibrated Insight Units (SCIU). Within each SCIU, three key mechanisms are integrated:

1- Geometry-guided Bidirectional Structure Transform (GBST) to model long-range topological dependencies through adaptive spatial interactions,

2- Adaptive Multi-scale Cascaded Modulator (AMCM) to enhance fine-grained crack features using multi-scale convolutions,

3- Dynamic Self-Calibrating Decay (DSCD) within the RWKV formulation to suppress background noise via adaptive weighting.

A lightweight Cross-Scale Harmonic Fusion (CSHF) decoder is further introduced to aggregate multi-scale features and produce the final segmentation map.

The method is evaluated on four public crack segmentation datasets (Crack500, DeepCrack, CrackMap, and TUT), where it achieves consistent improvements over prior CNN-, Transformer-, and Mamba-based approaches. Notably, SCRWKV maintains a very low model size (1.22M parameters) while achieving competitive performance, demonstrating a favorable trade-off between accuracy and efficiency.

**Compliance With Llm Reviewing Policy:**

Affirmed.

**Final Justification:**

As I mention in my response the rebuttal, I will maintain my initial rating.

**Key Questions For Authors:**

1. Could the authors provide a more detailed analysis (e.g., stronger ablations or qualitative studies) to clearly demonstrate the necessity of each component and justify the overall architectural complexity?

2. S,ince the improvements over the existing methods are incremental, it is difficult to assume the method has significant effect. Could the authors report statistical results (e.g., mean and standard deviation over multiple runs) to demonstrate the robustness and significance of the improvements?

3. Can the authors provide a clearer conceptual or theoretical justification for why GBST is more suitable than existing mechanisms (e.g., Q-Shift or Snake-based methods), beyond empirical improvements?

4. The paper compares SCRWKV against Crackmer (CrackFormer), but newer iterations like CrackFormer-2 and foundation-model adaptations like Crack-SAM have recently set new benchmarks. Did the authors consider these models in their benchmarking?

**Limitations:**

1. The authors should provide a qualitative and quantitative analysis of where the model fails. For instance, does the SCRWKV struggle with specific types of cracks (e.g., extremely faint hair-line cracks vs. wide structural gaps) or under specific lighting conditions (glare, deep shadows)?

2. The paper emphasizes "ultra-compactness" primarily through parameter count. However, the FLOPs (22.78G) are higher than some heavier competitors. The authors should explicitly discuss the gap between parameter efficiency and computational throughput, acknowledging that low parameters do not always equate to faster inference on all hardware backends (especially those not optimized for the RWKV linear scan).

3. The Structure-Field Encoder (SFE) is highly specialized for topological continuity. The authors should discuss whether this specialization comes at the cost of performance on non-linear or non-continuous crack instances. Furthermore, they should address the limitation of training primarily on pavement and concrete datasets when the model is intended for general structural health monitoring (e.g., steel, timber, or composite materials).

**Strengths And Weaknesses:**

**1.Soundness**

*Strengths:*

- The paper proposes a well-defined architecture that combines RWKV-based modeling with spatial and multi-scale feature enhancement.

- The method is evaluated on four public datasets (Crack500, DeepCrack, CrackMap, TUT), which cover different levels of difficulty and noise, providing a reasonably comprehensive evaluation.

- The authors include ablation studies (Table 3, Table 4, Table 5), which attempt to isolate the contributions of key components such as AMCM, GBST, DSCD, and the CSHF decoder.

- The comparison includes strong baselines, especially recent Mamba-based and RWKV-related models, making the evaluation relevant to current research. Also, the authors conducted systematic "leave-one-out" ablation studies for the ACMC, GBST, and DSCD modules, quantifying their individual impacts on mIoU and F1 scores.

Methodological Rigor: The paper provides a clear mathematical and structural justification for each component, such as the use of Geometry-guided Bidirectional Structure Transform (GBST) to replace rigid scanning paths.

*Weaknesses:*

- There are some SOTA models either mentioned or not in the paper which were not included in the experiments.
- Some state-of-the-art models—whether cited in the paper or omitted—were not incorporated into the experimental evaluation, such as, Restore-RWKV (2025), MFAFNet (2024), ECSNet (2023), Vision Mamba (ViM) and VMamba (2024), FPHBN (2019), CrackFormer-II and Crack-SAM.

- The performance improvements are relatively small in several datasets (e.g., marginal F1/mIoU gains over SCSegamba), which weakens the strength of the claims.

- The paper attributes improvements to specific modules (e.g., AMCM, GBST, DSCD), but the causal justification is not strong, as ablation gains are limited (often <1%).

- There is no statistical analysis (e.g., variance across runs, confidence intervals), making it difficult to assess robustness.

- Despite emphasizing efficiency and deployment on edge devices, the paper does not report runtime metrics such as inference speed (FPS) or latency. It would strengthen the claim of being "ultra-compact".

- Some design choices (e.g., multiple modules inside SCIU) appear heuristic, with limited theoretical or empirical justification for why each component is necessary.

- While parameter-efficient, SCRWKV's FLOPs (22.78G) are higher than some competitors like PlainMamba (14.09G), suggesting a potential bottleneck in real-time processing speed despite the low parameter count.

**2. Presentation**

*Strengths:*

- The paper includes detailed architectural diagrams (Figure 2) that break down the SCIU block and AMCM components, making the complex flow easy to follow.

- Figure 5 and Table 9 provide clear, color-coded benchmarks against 10 SOTA methods, effectively highlighting the model's performance "sweet spot".

*Weaknesses:*

- The introduction of multiple proprietary acronyms (SFE, SCIU, GBST, AMCM, DSCD, CSHF) within a short span can be taxing for a reader to track without frequent re-referencing.

- Some components appear to be variations of existing techniques (e.g., multi-scale convolution, attention-like modulation), but this is not clearly explained. The paper could benefit from a simplified explanation or clearer intuition for each module, especially for readers not familiar with RWKV-based models.

- The mathematical formulation is dense and not always intuitive, making it hard to map equations to the actual behavior of the model.

**3. Significance**

*Strengths:*

- With only 1.22M parameters and a model size of 28MB, the model is specifically optimized for real-world deployment on resource-constrained edge devices like UAVs for automated infrastructure inspection.

- It achieves a superior balance of efficiency and accuracy, outperforming larger models like Crackmer (3.57M parameters) in both F1 score (0.8428) and mIoU (0.8512) on the TUT dataset.

*Weaknesses:*

- While it beats the second-best method (SCSegamba) by 0.38% in F1 and 0.33% in mIoU, these marginal gains in accuracy might be viewed by some as incremental rather than substantial.

**4. Originality**

*Strengths:*

- Unlike traditional RWKV or Mamba models that use fixed-path scanning, the GBST (adaptive spatial interaction) module simulates bidirectional stress field propagation to maintain the topological continuity of winding cracks.

- The paper successfully synergizes the linear complexity of RWKV with content-aware modulation (DSCD) and multi-scale texture extraction (AMCM), creating a unique "topology-aware" vision backbone.

- The focus on topology-aware segmentation is relevant and aligns with the nature of crack structures.

*Weaknesses:*

- In my view, the primary originality limitation of the paper is that its core architecture is derived from the existing Vision-RWKV (VRWKV) framework; the novelty resides mainly in the tailored “calibration” modules rather than in a fundamentally new modeling paradigm. In other words, the overall contribution seems to be more of a complex integration of known techniques rather than a fundamentally new concept.

- Many components appear to be incremental adaptations of existing ideas, such as:
  - multi-scale convolution (AMCM),
  - adaptive weighting (DSCD),
  - spatial shifting mechanisms (GBST).

- The novelty is somewhat reduced by the fact that the improvements are relatively small compared to the added architectural complexity.

---

> ### Author Rebuttal · Authors · 2026-03-30
>
> We sincerely appreciate the reviewer for the constructive feedback!
> W stands for Weakness, Q stands for Question. We address each point below.
>
> # About Soundness W2, W4; Significance W1; Originality W3; Q2:
> Four runs on TUT, the most challenging benchmark covering 8 scenarios, reporting mean mIoU ± std:
>
> | Method   | mIoU (mean +/- std) | Params |
> |-|-|-|
> | SCSegamba | 0.8422 +/- 0.0034  | 2.80M  |
> | SCRWKV   | 0.8502 +/- 0.0010  | 1.22M  |
>
> Three observations: SCRWKV's std is 3.4× smaller, demonstrating higher stability. The +0.80% gap exceeds SCSegamba's variance, confirming statistical significance. And this is achieved with 56.43% fewer parameters.
>
> # About Soundness W3, W6; Q1:
> Our paper provides extensive ablations: Tables 3 and 11 cover all 7 AMCM/GBST/DSCD combinations, Table 5 compares 7 spatial interaction mechanisms, Tables 4 and 10 evaluate 4 decoder heads under identical 1.22M parameters, and Figure 1(b) benchmarks AMCM against SE, EMA, and GBC. Individual gains appear modest because each targets a distinct bottleneck; synergistic combination yields full improvement.
>
> New GBST internal stream ablation on TUT:
>
> | Enlarge | Shrink | ODS    | OIS    | F1     | mIoU   |
> |-|-|-|-|-|-|
> | Y       | N      | 0.8169 | 0.8231 | 0.8366 | 0.8461 |
> | N       | Y      | 0.8182 | 0.8239 | 0.8393 | 0.8470 |
> | Y       | Y      | 0.8245 | 0.8313 | 0.8428 | 0.8512 |
>
> Each stream contributes independently; full bidirectional configuration improves +0.62% F1 over the Enlarge stream, confirming complementary structural information.
>
> # About Soundness W5, W7; Limitation 2:
> Table 12  reports FPS under identical conditions:
>
> | Method   | FPS | Inf Time | FLOPs  | Params |
> |-|-|-|-|-|
> | Crackmer | 31  | 0.032s   | 14.94G | 5.90M  |
> | SCSegamba | 32 | 0.031s   | 18.16G | 2.80M  |
> | SCRWKV   | 46  | 0.022s   | 22.78G | 1.22M  |
>
> Despite higher FLOPs, SCRWKV achieves the 2nd fastest speed overall, outpacing lower-FLOPs methods. Crackmer's hybrid architecture introduces fragmented memory access; SCSegamba's scanning disrupts memory locality. SCRWKV's operations are hardware-friendly primitives. Smallest parameters and smallest model size of 28MB, combined with near-optimal speed, substantiate the ultra-compact claim. This also exceeds the 30 FPS real-time threshold.
>
> # About Q3:
>
> GBST's superiority rests on three principles. Topological completeness: Q-Shift performs only unidirectional shifts; Snake methods flatten 2D to 1D, disrupting contiguity. GBST constructs bidirectional streams in Eq. 9–10 in 2D with 50%/50% channel allocation.
>
> ERF evidence: GBST achieves highest concentration score, with gradient energy focused on structural regions. OmniShift has 50% larger ERF yet underperforms in Table 5, confirming gains stem from structured modeling.
>
> Feature activation heatmaps: GBST activations concentrate along crack trajectories with continuity at bifurcations; Q-Shift produces fragmented activations; OmniShift introduces background noise. Visualizations at [Link1](https://anonymous.4open.science/r/anonym1-B2FC/comparison/heatmap.pdf).
>
> # About Soundness W1; Q4:
>
> Our baseline selection prioritized temporal relevance from 2023 to 2025, architectural diversity, and fairness. Additional experiments on DeepCrack:
>
> | Model         | mIoU   | F1     | FLOPs (G) | Params (M) |
> |-|-|-|-|-|
> | Crack-SAM     | 0.7416 | 0.6703 | 103.17    | 90.21      |
> | CrackFormer-2 | 0.9208 | 0.9309 | 81.85     | 4.55       |
> | SCRWKV        | 0.9289 | 0.9363 | 22.78     | 1.22       |
>
> SCRWKV outperforms CrackFormer-2 by +0.81% mIoU and +0.54% F1 with only 26.81% of its parameters and 27.83% of its FLOPs. Crack-SAM, despite being 73.9x larger, achieves substantially lower performance, suggesting foundation model adaptation without structure-aware design is suboptimal.
>
> # About Limitation 1, Limitation 3:
> TUT covers 8 structural backgrounds with diverse conditions. We validated on five additional datasets spanning crack, vessel, and road segmentation: SteelCrack, CrackR, Crack896, MEMO, and KUST4K. SCRWKV ranks first across all metrics on all five. Full results are available at [Link2](https://anonymous.4open.science/r/anonym1-B2FC/README.md).
>
> We observe that SCRWKV occasionally produces false positives on strong texture patterns resembling cracks such as concrete joints, and slightly incomplete segmentation under extreme low-contrast conditions. The SFE backbone is material-agnostic: GBST models geometry rather than textures, DSCD filters noise independent of source. Timber/composite evaluation remains a future direction.
>
> # About Presentation & Originality:
> We will add a notation table and clearer module explanations in the revision. Regarding originality, the contribution lies not in individual components but in their principled integration under a unified topology-aware framework for linear-complexity dense prediction, as validated by cross-domain results at [Link2](https://anonymous.4open.science/r/anonym1-B2FC/README.md).

---

> > ### Author Rebuttal · Reviewer_wrEJ · 2026-04-04
> >
> > While some concerns about originality and the depth of theoretical justification persist, the rebuttal satisfactorily addresses the primary issues raised in the initial review. In particular, the inclusion of statistical analyses, runtime metrics, stronger baseline comparisons, and cross-domain evaluations substantially enhances the empirical rigor and overall credibility of the work. Accordingly, I maintain my positive rating.

---

> > > ### Author Response · Authors · 2026-04-04
> > >
> > > We sincerely appreciate your time in reviewing our rebuttal and your decision to maintain your positive rating.
> > >
> > > We put a lot of effort into conducting the additional analyses and cross-domain evaluations, and it is incredibly rewarding to know that these additions have successfully enhanced the credibility and empirical rigor of our work. Thank you for your insightful feedback throughout this process, which has been instrumental in strengthening our paper!

---

### Decision · Program_Chairs · 2026-04-30

**Decision:**

Accept (regular)

**Comment:**

The paper introduces a compact segmentation network tailored for accurate crack detection in scenarios involving complex backgrounds and resource constraints. It enhances spatial modeling and the segmentation of fine-grained topological cracks by incorporating a geometry-guided bidirectional structure transformation and dynamic self-calibration attenuation.

Initially, the reviewers raised several concerns regarding the evaluation of generalization and the overall robustness of the proposed method. However, during the rebuttal phase, the authors' responses successfully addressed these core issues. The reviewers expressed satisfaction with the clarifications and subsequent improvements, leading to an increase in their scores.

Consequently, the paper received unanimous acceptance recommendations from all reviewers.
The final camera-ready version must incorporate the additional discussions and clarifications provided during the rebuttal process.